



# Lagrangian condensation microphysics with Twomey CCN activation

Wojciech W. Grabowski[1,2], Piotr Dziekan[2], and Hanna Pawlowska[2]

[1]National Center for Atmospheric Research, Boulder, CO, USA
[2]Institute of Geophysics, Faculty of Physics, University of Warsaw, Warsaw, Poland

*Correspondence to:* Wojciech W. Grabowski (grabow@ucar.edu)

**Abstract.**

We report the development of a novel Lagrangian microphysics methodology for improved simulations of warm ice-free clouds. The approach applies the traditional Eulerian method for the momentum and continuous thermodynamic fields, the temperature and water vapor mixing ratio, and uses Lagrangian "super-droplets" to represent condensed phase such as cloud

droplets and drizzle/rain drops. In all other applications of the Lagrangian warm-rain microphysics, the super-droplets outside clouds represent un-activated cloud condensation nuclei (CCN) that become activated upon entering a cloud and can further grow through diffusional and collisional processes. The original methodology allows studying in detail not only effects of CCN on cloud microphysics and dynamics, but also CCN processing by a cloud. However, when cloud processing is not of interest, a simpler and computationally more efficient approach can be used with super-droplets forming only when CCN is

activated and no super-droplet existing outside a cloud. This is possible by applying the Twomey activation scheme where the local supersaturation dictates the concentration of cloud droplets that need to be present inside a cloudy volume, as typically used in Eulerian bin microphysics schemes. Since a cloud volume is a small fraction of the computational domain volume, the Twomey super-droplets provide significant computational advantage when compared to the original super-droplet methodology. Additional advantage comes from significantly longer time steps that can be used when modeling of CCN deliquescence

is avoided. Moreover, other formulation of the droplet activation can be applied in case of low vertical resolution of the host model, for instance, linking the concentration of activated cloud droplets to the local updraft speed.

This paper discusses the development and testing of the Twomey super-droplet methodology focusing on the activation and diffusional growth. Details of the activation implementation, transport of SDs in the physical space, and the coupling between super-droplets and the Eulerian temperature and water vapor field are discussed in detail. Some of these are relevant to the

original super-droplet methodology as well and to the ice phase modeling using the Lagrangian approach. As a computational example, the scheme is applied to an idealized moist thermal rising in a stratified environment, with the original super-droplet methodology providing benchmark to which the new scheme is compared.



## 1 Introduction

Traditional cloud modeling methodologies apply continuous medium approach for all thermodynamic variables, that is, not only for the temperature and water vapor, but also for all forms of cloud condensate and precipitation. Such methodologies have been the workhorse of the cloud-scale modeling from its early days (e.g., Kessler, 1963; Liu and Orville, 1969; Murray, 1970;

Schlesinger, 1973; Klemp and Wilhelmson, 1978; Clark, 1979) , but also in numerical weather prediction using global as well as limited-area models and in climate simulation. Since the edge of an ice-free cloud represents a sharp transition from droplet-laden air close to saturation to unsaturated droplet-free air outside the cloud, numerical diffusion and dispersion errors impose stringent constraints on numerical schemes suitable for cloud modeling. For instance, since cloud and precipitation variables are positive definite, any numerical scheme that introduces negative values to the numerical solution (e.g., during advection in

the physical space) is not suitable for cloud simulation. Moreover, difficulties in representing sharp cloud edge discontinuities in thermodynamic fields are well appreciated by the cloud-scale modeling community, especially from the point of view of the supersaturation field, the key variable for the formation and growth of water and ice cloud particles (e.g. Grabowski and Morrison, 2008, and references therein).

The last couple decades witness an increased interest in cloud-scale computational approaches that limit the abovementioned

problems and attempt to better represent the truly multiphase nature of clouds. Among those, the particle-based Lagrangian method, referred to as the Lagrangian Cloud Model (Andrejczuk et al., 2008, 2010) or the "super-droplet method" (Shima et al., 2009) , is of particular relevance (see also Riechelmann et al., 2012; Arabas et al., 2015; Hoffmann et al., 2015, among others) . By representing formation and growth of natural cloud particles using a subset of those particles ("super-particles") many problem haunting traditional Eulerian approaches are either eliminated or significantly reduced. For instance, formation

of cloud droplets through activation of cloud condensation nuclei (CCN) can be formulated in a straightforward way and processing of CCN though collision/coalescence or chemical reactions within droplets can be simulated from first principles. In the continuous medium approach, on the other hand, these processes require either extreme computational effort (i.e., multidimensional bin schemes) or are simply impossible to consider without additional simplifications. In the Lagrangian approach for warm (ice-free) clouds, each super-droplet carries a set of attributes (such as CCN size and composition, wet

particle mass and multiplicity, etc.) that allow representing condensation and associated latent heat release as well as the development of drizzle and rain. In previous applications of such a methodology, the super-droplets outside clouds represent un-activated CCN (haze) particles that become activated upon entering a cloud and can further grow through diffusional and collisional processes. Since the information about the CCN is available for each super-droplet, the methodology allows studying in detail not only effects of CCN on cloud microphysics and dynamics, but also CCN processing by a cloud. However, when

cloud processing is of no interest, the Twomey activation can be used with super-droplets forming when CCN is activated and no super-droplet existing outside a cloud (e.g., Grabowski et al., 2011), as often applied in Eulerian bin microphysics models. Since cloud volume is a small fraction of the computational domain volume, the Twomey super-droplets allow significant savings when compared to CCN-based Lagrangian methodology. Moreover, significantly longer time steps can be used because modeling of CCN deliquescence is avoided.





This paper discusses development and testing of a novel Lagrangian approach focusing on activation and diffusional growth of cloud droplets. Our motivation is to use this methodology to study the impact of turbulence and entrainment on the spectrum of cloud droplets in shallow warm boundary layer clouds, such as tropical or subtropical cumulus and subtropical stratocumulus (see idealized adiabatic parcel simulations discussed in Grabowski and Abade, 2017). The key aspect, difficult if not impossible to apply in the Eulerian approach, is the possibility to formulate a subgrid-scale statistical scheme and apply it to individual droplets taking advantage of a stochastic formulation along the Lagrangian particle trajectory as in Grabowski and Abade (2017). The developments discussed here exclude collision/coalescence as only marginally relevant to the spectral broadening problem. Collision/coalescence can be included in a relatively straightforward way (see a review and tests of various approaches discussed in Unterstrasser et al., 2017) and adding it to the model described here will be pursued in the future.

The next section presents analytic formulation of the Twomey super-droplet scheme and discusses its implementation in the Eulerian fluid flow model. The specific aspects discussed in detail are the treatment of the activation on the finite-difference fluid flow model grid, transport of super-droplets across the Eulerian grid, and coupling between the super-droplets and Eulerian thermodynamics. Section 3 presents examples of model simulations where the Lagrangian thermodynamics is included in an anelastic small-scale fluid flow model and applied in moist rising thermal simulations. A traditional super-droplet scheme (i.e., following CCN particles and allowing their activation and growth of resulting cloud droplets) is used to show consistency between the two methods. Brief conclusions and the outlook are presented in section 4.

## 2 Formulation

### 2.1 Analytic formulation

Model equations describe evolutions in space and time of the potential temperature, water vapor mixing ratio, and of a set of Lagrangian point particles representing activated cloud droplets. The potential temperature $\Theta$ and water vapor mixing ratio $q_v$ equations are:

$$\frac{\mathrm{D}\Theta}{\mathrm{D}t} = \frac{L_v}{c_p \Pi} C_d \,, \tag{1}$$

$$\frac{\mathrm{D}q_v}{\mathrm{D}t} = -C_d \,, \tag{2}$$

where $\mathrm{D}/\mathrm{D}t = \partial/\partial t + (\boldsymbol{u} \cdot \nabla)$ is the material (advective) derivative, $C_d$ is the condensation rate, $\Pi = (p/p_0)^{R/c_p}$ is the Exner function ($p$ is the local pressure that in the anelastic system comes from the environmental profiles and $p_o = 1000$ hPa), and $L_v$ and $c_p$ are the latent heat of vaporization and air specific heat at constant pressure, respectively. The condensation rate $C_d$ is



defined as the rate of change of the mass of cloud droplets. For the finite-difference model considered here, it can be calculated from the rate of change of mass of all cloud droplets located within a given grid cell:

$$C_d = \frac{\mathrm{D}}{\mathrm{D}t} \left( \sum_{i=1}^{N} \frac{4\rho_w}{3\rho_a} \pi r_i^3 N_i \right) , \tag{3}$$

where $\rho_w$ and $\rho_a$ are the water and air density, respectively, $r_i$ and $N_i$ are the radius and concentration of $N$ cloud droplet classes (bins) into which all droplets located within the grid cell are grouped. Such a definition has some similarity to the way condensation rate is calculated in Eulerian bin microphysics schemes, an analogy that will be useful when droplet activation is discussed later in this section. Given the supersaturation $S = q_v/q_{vs} - 1$ (where $q_{vs}$ is the saturated water vapor mixing ratio) the individual droplet growth equation is:

$$\frac{\mathrm{d}r_i}{\mathrm{d}t} = \frac{AS}{r_i r_0} , \quad \text{where} \quad A = \frac{q_{vs} D_v}{\rho_w \left( 1 + \frac{L_v}{c_p} \frac{\partial q_{vs}}{\partial T} \right)} , \tag{4}$$

$r_0 = 1.86$ μm is a parameter that allows including kinetic effects (e.g., Clark, 1974; Kogan, 1991) and $D_v$ is the diffusivity of water vapor in the air that depends on the temperature and pressure. A convenient feature of (4) is that the rate of growth remains bounded when $r_i$ approaches zero. The coefficient $A$ used in (4) is an approximate form of a more general formulation as given, for instance, by Eq. 3 in Grabowski et al. (2011). The approximate formulation (4) can be obtained by assuming that the thermal conductivity of air $K$ is approximately given by $K = c_p \rho_a D_v$. Note that Grabowski et al. (2011) and Grabowski and Abade (2017) applied a constant value $A = 0.9152 \times 10^{-10}$ m$^2$s$^{-1}$. Droplets are carried by the airflow (i.e., droplet sedimentation is excluded), an assumption justifiable by the exclusion of droplet collisions, spatial scales considered (tens of meters and larger), and the length of simulations (up to a few tens of minutes). Thus, the evolution of the $i$th droplet position $\boldsymbol{x}_i$ is calculated as

$$\frac{\mathrm{d}\boldsymbol{x}_i}{\mathrm{d}t} = \boldsymbol{u}(\boldsymbol{x}_i, t) , \tag{5}$$

where $\boldsymbol{u}$ is the air flow velocity predicted by the dynamical model.

Considering typical cloud droplet concentrations in natural clouds, from several tens to a few thousands per cubic centimeter, it is computationally impossible to follow all cloud droplets in the entire volume of even a very small cloud. Thus, the Lagrangian methodology involves following only a selected (typically relatively small) subset of cloud droplets, referred to as super-droplets following Shima et al. (2009). This is again in the spirit of using a finite (and typically relatively small) number of classes (bins) in the Eulerian bin microphysics scheme. Note that since each super-droplet represents a multiplicity of real cloud droplets, the super-droplet position has no clear physical interpretation. Arguably, the position corresponds to the location of one of the droplets that the super-droplet represents. The location is needed to assign the super-droplet to a particular grid cell of the Eulerian fluid flow model and to represent advection in the physical space as given by (5). A usual



interpretation is that a super-droplet represents all droplets located at a given moment in a particular grid cell. However, such an interpretation leads to a conceptual inconsistency because physically moving the super-droplet from one grid cell to another does not mean that all droplets that the particular super-droplet represents do the same. It is not clear if a physically consistent and computational efficient methodology can be developed to avoid this conceptual problem.

5    As in Andrejczuk et al. (2008, 2010); Shima et al. (2009) and Riechelmann et al. (2012) among others, the list of attributes for each super-droplet includes the position $x_i$, radius $r_i$, and multiplicity. The latter depicts the number of particles represented by a single super-droplet. In the numerical implementation, the multiplicity is assumed to be the number mixing ratio, that is, $N_i/\rho_a$, similarly as in Eulerian bin microphysics schemes. Other attributes can be added if needed, for instance, the local supersaturation perturbation (on top of the grid-scale supersaturation predicted by the flow model) that can affect super-droplet growth in (4) as in Grabowski and Abade (2017) or the subgrid-scale velocity perturbation that can affect the motion of the super-droplet in (5).

## 2.2 Numerical implementation

As will be discussed in section 3, the novel super-droplet scheme has been included into the finite-difference anelastic model EULAG and its simplified version referred to as babyEULAG. EULAG and babyEULAG apply nonoscillatory-forward-in-time (NFT) integration scheme (e.g., Smolarkiewicz and Margolin, 1993; Grabowski and Smolarkiewicz, 2002; Prusa et al., 2008). For the coupling with super-droplets, the NFT scheme for the potential temperature (1) and water vapor mixing ratio (2) has been modified to include the Euler-forward time integration, that is,

$$\Psi(t + \Delta t) = [\Psi(t) + F(t)\,\Delta t]_0 , \qquad (6)$$

where $\Psi$ is either $\Theta$ or $q_v$, $F$ represents the right-hand-side of (1) and (2), and subscript "0" depicts the departure point of the fluid trajectory. This is the same as applied in the bin-microphysics versions of EULAG in Wyszogrodzki et al. (2011, section 2.2 therein) and babyEULAG in Grabowski and Jarecka (2015, appendix therein). Exploring the analogy between Lagrangian (trajectory-wise) and Eulerian (control-volume-wise) description of the fluid flow equations, (6) is solved using the flux-form monotone advection scheme MPDATA (e.g., Smolarkiewicz, 2006). Thus, the second-order-in-space and centered-in-time advection scheme is combined with the first-order-in-time (Euler forward) integration of the forcing term. Similar approach is used for the super-droplets, where the super-droplet transport is computed using the predictor-corrector scheme and droplet growth is calculated using the first-order-in-time un-centered scheme. It should be stressed the momentum equation in the host babyEULAG and EULAG models is advanced applying the centered in time scheme.

## 2.3 Super-droplet initiation

The key element of the scheme presented here that makes it distinct from the approach used in (Andrejczuk et al., 2008, 2010), Shima et al. (2009), Riechelmann et al. (2012), Arabas et al. (2015) and others, is the way super-droplets are created. The original implementations assume that super-droplets fill the entire computational domain, and they initially represent



deliquesced (humidified) CCN in equilibrium with their local environment. These un-activated super-droplets may become activated if environmental conditions dictate so, for instance, when passing through the cloud base. When CCN dry radius is one of the super-droplet attributes, the original approach allows explicit representation of aerosol processing by a cloud when collision/coalescence takes place (in which case the dry CCN after collision/coalescence combines dry CCN from col-

liding droplets; Shima et al., 2009) or when chemical reactions are included (e.g., A. Jaruga; PhD dissertation, University of Warsaw). However, if neither of those processes is of interest, a significantly simpler approach can be used based on the so-called Twomey activation (Twomey, 1959) as often used in bin microphysics schemes (e.g., Grabowski et al., 2011). Twomey approach links the number mixing ratio of activated CCN $N$ to the maximum supersaturation $S$ experienced by the cloudy volume. We will refer to the analytical or tabulated correspondence between $N$ and $S$ as the Twomey relationship. Cloud base

activation in the Eulerian bin microphysics scheme is simulated by introducing cloud droplets into appropriate bins until the supersaturation reaches its peak and activation is completed (see Grabowski et al., 2011). Without collision/coalescence, local droplet number mixing ratio provides information about the maximum supersaturation experienced by the volume in the past. With collision/coalescence, additional model variable, the number mixing ratio of already activated CCN, needs to be used to control whether additional CCN activation is required (see section 2c in Morrison and Grabowski, 2008) . Similarly, the

additional variable is needed if a significant variability of the CCN exists in the computational domain (e.g., in the vertical direction).

The same approach can be used with super-droplets as already applied in Grabowski and Abade (2017) (hereafter GA17) in adiabatic parcel simulations. The key idea is that super-droplets are created in supersaturated conditions when the local concentration of activated droplets as given by the Twomey relationship is smaller than the one dictated by the local supersaturation.

When a complete evaporation of cloud droplets occurs in sub-saturated conditions, super-droplets are simply removed from the super-droplet list. Hence, no super-droplets exist outside of cloudy volumes, similarly to traditional Eulerian bin microphysics schemes. It follows that super-droplets with Twomey activation provide significant computational advantage over the traditional Lagrangian approach because only a relatively small number of super-droplets has to be used. Note that in the Eulerian bin scheme the computational expense of the droplet transport in the physical space is independent of whether droplets fill a

small or a large fraction of the domain. This is because each bin needs to be advected separately in the physical space and the computational effort is independent of whether the entire domain or just its small fraction is filled with droplets.

We assume the same CCN characteristics as in GA17 and Arabas et al. (2015). CCN characteristics include the chemical composition, the number mixing ratio of activated CCN for a given supersaturation (the Twomey relationship) and the activation radius. CCN are assumed to be composed of sodium chloride (sea salt; NaCl). Idealized CCN distribution, the same as in Arabas

et al. (2015), is represented by a sum of two lognormal distributions with number mixing ratio, mean radii, and geometric standard deviations (unitless) as 57.33 and 38.22 $\mathrm{mg^{-1}}$ [i.e., per $\mathrm{cm^{-3}}$ for the air density of $1\,\mathrm{kg\,m^{-3}}$; these values come from converting the 60 and 40 $\mathrm{cm^{-3}}$ concentrations to the number mixing ratio using air density at the bottom of the computational domain in simulations discussed in section 3]; 20 and 75 nm; and 1.4 and 1.6, respectively. The N-S relationship is tabulated and the table is used as input to the super-droplet scheme. Once activated, the initial radius corresponding to the activation

radius is assigned for each super-droplet. The latter is approximated as $8 \times 10^{-10}/S_{\mathrm{act}}$ [m] as in GA17, where $S_{\mathrm{act}}$ is the

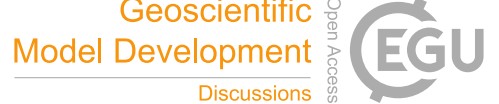



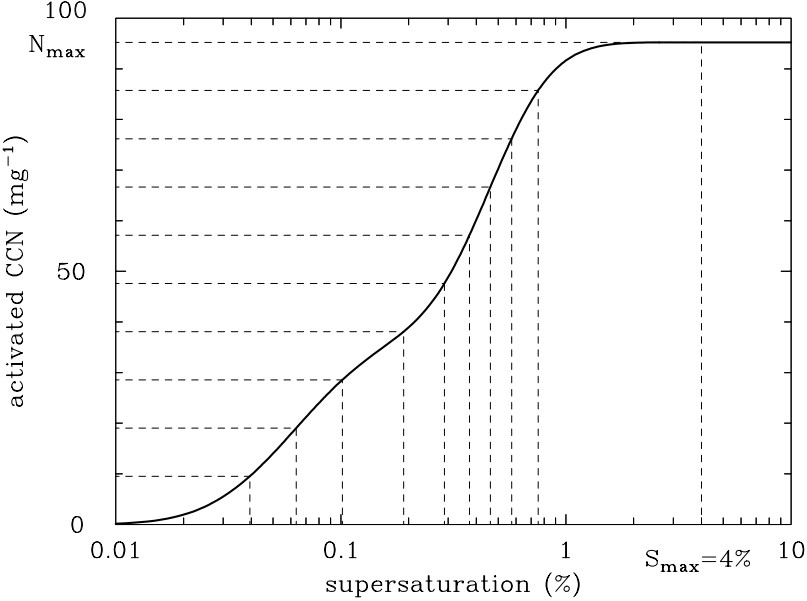

**Figure 1.** Thick line: number mixing ratio of activated CCN as a function of the supersaturation, the Twomey relationship, used in simulations described herein. Thin dashed lines illustrate numerical implementation of the CCN activation scheme. See text for details.

activation supersaturation; see Eq. (6) and Fig. 2 in Grabowski et al. (2011). In addition to the droplet radius, the model keeps track of the super-droplet number mixing ratio (i.e., the number of droplets per unit mass of dry air) that corresponds to the multiplicity parameter (or attribute) of Shima et al. (2009). A newly created super-droplet is placed randomly within a given grid cell and added to the super-droplet list.

Figure 1, adopted from GA17, shows the N-S relationship and illustrates the way super-droplets are created. First, the maximum supersaturation $S_{max}$ is selected. $S_{max}$ has to exceed the maximum supersaturation anticipated in the simulation. $S_{max}$ equal to 4% is used here as shown in the figure. The corresponding maximum number mixing ratio of activated droplets $N_{max}$ is divided by the number of droplet classes to be used in the simulations. The example in Fig. 1 assumes 10 classes whereas simulations typically apply several tens to several thousands classes. New super-droplets are introduced to a given

grid cell when the supersaturation predicted for that grid cell exceeds the supersaturation corresponding to the activation supersaturation of super-droplets already present in the grid cell. The approach illustrated in Fig. 1 ensures that the multiplicity parameter is the same for all super-droplets. This is beneficial because equal multiplicity minimizes statistical fluctuations of derived cloud quantities (such as the droplet concentration or liquid water content) when super-droplets are advected from one grid cell to another. The approach adopted here was suggested by simple one-dimensional advection tests completed during

early stages of the scheme development.

When applied in a multidimensional fluid flow model, there is an additional issue with the proposed scheme that needs to be addressed. Figure 2 shows a single two-dimensional grid cell at which formation of new super-droplets takes place at time $t$. At the next time step, $t + \Delta t$, super-droplets are advected upwards by the updraft, and a droplet-free volume is advected



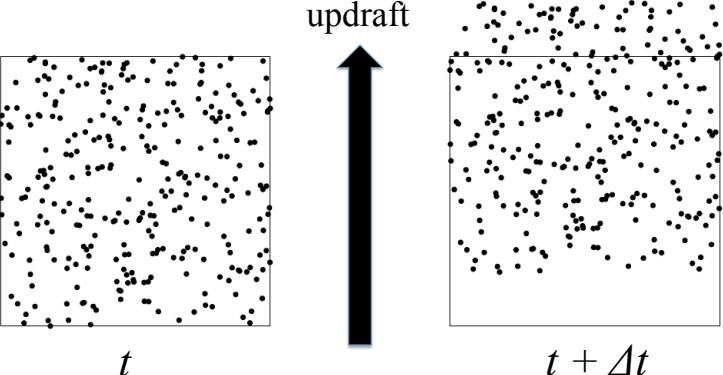

**Figure 2.** Illustration of the activation as represented on the fluid flow grid. Left panel shows locations of CCN activated at a given model time step. Right panel shows the situation at the next time step when activated CCN are advected away from the grid cell and activation of new CCN is required.

into the grid cell. Assuming that the supersaturation within the grid cell does not change, there is a need to activated new super-droplets as some of those present within the grid cell at the previous time step moved upwards. The new super-droplets should be introduced into the droplet-free volume (i.e., in the lower part of the grid cell in the right panel of Fig. 2) because un-activated CCN would be there. However, keeping track of volumes void of super-droplets during activation followed by

advection is cumbersome. At the same time, adding new super-droplets randomly into the entire grid cell leads to the situation where super-droplets are not randomly distributed (i.e., more super-droplets is present in the upper part of the grid cell in Fig. 2). A simple approach adopted here is that all super-droplets are always randomly repositioned within a given grid cell once additional activation within that cell takes place.

### 2.4   Transport of super-droplets in the physical space

Super-droplets are advected in the physical space applying a predictor-corrector scheme to solve Eq. 5. The predictor step estimates the n+1 time level position from n time level velocity as:

$$\boldsymbol{x}_p^{n+1} = \boldsymbol{x}^n + \boldsymbol{u}^n(\boldsymbol{x}^n)\Delta t \,. \tag{7}$$

where the subscript "p" depicts the predictor solution. The corrector step (subscript "c") is subsequently applied as:

$$\boldsymbol{x}_c^{n+1} = \boldsymbol{x}^n + \left[\boldsymbol{u}^{n+1}(\boldsymbol{x}_p^{n+1}) + \boldsymbol{u}^n(\boldsymbol{x}^n)\right] \frac{\Delta t}{2} \,. \tag{8}$$

The predictor-corrector scheme ensures the second-order accuracy for the time integration of the super-droplet transport. However, to increase accuracy, the corrector step can be repeated by replacing $\boldsymbol{x}_p$ by already calculated $\boldsymbol{x}_c$ in the $\boldsymbol{u}^{n+1}$





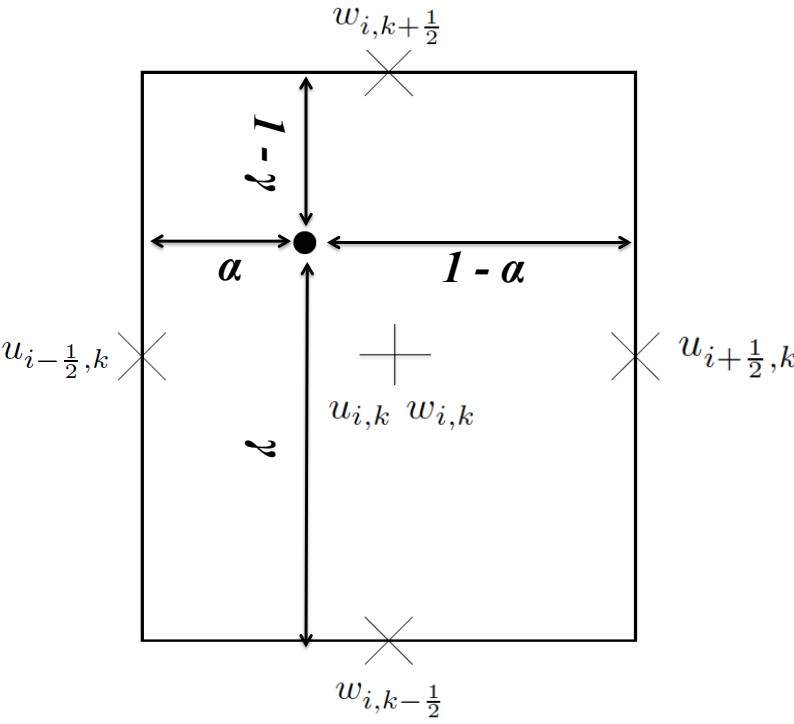

**Figure 3.** Illustration of the interpolation scheme used in the super-droplet transport scheme referred to as 'simple' in the text. The rectangular box represents a single grid cell with u and w depicting horizontal and vertical velocities perpendicular to grid cell boundaries used in the advection scheme of the Eulerian model. The large dot represents droplet position.

velocity on the right hand side of (7). Note that velocity needs to be interpolated to the super-droplet position and repeating the corrector step increases the oveall computational cost. We will test the benefit of the second correction step in the droplet advection procedure later in this section. It also needs to be pointed out that the super-droplet transport requires knowledge of the flow velocity at the $n+1$ time level in (7). Similarly to the case of EULAG's and babyEULAG's advection of the temperature and water vapor mixing ratio where advecting velocities need to be known at $n+1/2$ time level, the $n+1$ time level velocities in (8) are extrapolated from velocities avilable at $n-1$ and $n$ time levels.

Velocity interpolation to calculate super-droplet transport is the key element of the Lagrangian scheme. Since the EULAG model applies un-staggered grid (i.e., all variables are located at the same position), one possibility is to consider a grid cell whose 4 corners in 2D (8 vertices in 3D) form a rectangular (cuboid-shaped in 3D) grid cell. For a super-droplet located in such a grid cell, flow velocity at the droplet position can be interpolated from the velocity values at the corners/vertices. Arguably the simplest possibility is to apply a bi-linear (tri-linear in 3D) interpolation scheme, but more advanced scheme may be considered as well. However, the bi-linear interpolation (and likely more advanced interpolation schemes) does not lead to physically consistent results as documented below.





Advection of the potential temperature and water vapor mixing ratio (as well as the velocity components) in EULAG is performed on the C grid (i.e., with the horizontal/vertical velocities at the vertical/horizontal grid cell boundaries). Advective velocities come from interpolating velocity components predicted on the un-staggered grid into the C grid. Advective velocities satisfy the anelastic incompressibility condition $\nabla \cdot (\rho \boldsymbol{u}) = 0$, where $\rho(z)$ is the anelastic density profile. In 2D, the divergence

of advecting velocities can be written in the finite-difference form as (see Fig. 3):

$$\frac{u_{i+\frac{1}{2},k} - u_{i-\frac{1}{2},k}}{\Delta x} + \frac{w_{i,k+\frac{1}{2}} - w_{i,k-\frac{1}{2}}}{\Delta z} = -\frac{w}{\rho}\frac{\partial \rho}{\partial z}, \tag{9}$$

where the term on the right hand side of (9) representing the change of the anelastic density with height is left in the analytic form as irrelevant to the discussion. With a single super-droplet located in the grid cell (see Fig. 3), the horizontal and vertical velocities can be interpolated using a simple scheme similar to that used in Arabas et al. (2015):

$$u = \alpha u_{i+\frac{1}{2},k} + (1-\alpha)u_{i-\frac{1}{2},k}, $$
$$w = \gamma w_{i,k+\frac{1}{2}} + (1-\gamma)w_{i,k-\frac{1}{2}}, \tag{10}$$

where $\alpha$ and $\gamma$ are nondimensional distances of the super-droplet position to the cell boundary as shown in Fig. 3. As documented in the Appendix 1, such a definition ensures that the incompressibility condition (9) is maintained on the subgrid-scale of the grid cell. This, on the other hand, ensures that a deformation of the initially rectangular grid cell as represented by passive advection of all passive partricles initially located inside the cell preserves the cell area (volume in 3D). We refer to the

interpolation scheme (10) as "simple" in the following discussion in contrast to the bi-linear (or tri-linear in 3D) interpolation scheme introduced previously.

To investigate accuracy of the super-droplet transport scheme, a relatively simple test problem was designed. In the test, 2D rising moist thermal simulations driven by the Eulerian condensation bulk scheme were used applying the same simulation setup as in the super-droplet simulations (see section 3.1). The predicted rising thermal flow (similar to the one shown later

in the paper applying super-droplets) was applied to advect a large number of passive particles introduced to a fraction of the computational domain including the thermal and its immediate environment at the onset of the simulation. The number of passive particles varied from several tens to a few thousands per grid cell in various tests. In the rising thermal flow simulated by the model, one should expect the average number of particles per grid volume to slightly decrease because of the density decreasing with height. Moreover, the number should show statistical fluctuations due to advection of particles from one grid

cell to another. The fluctuation amplitude should vary approximately as an inverse of the square root of the initial number of particles per grid cell. These assumptions provide the basis for evaluating the accuracy of the super-droplet transport.

Figure 4 shows evolutions of the minimum and maximum number of passive super-droplets per grid cell advected using the predictor-corrector scheme with the upper and lower panels showing results from the bi-linear and simple flow velocity interpolation, respectively. The extrema are calculated using only grid cells with the cloud water mixing larger than 0.01 g/kg.

Results are shown from simulations applying the predictor only scheme (8), the predictor-corrector scheme (8) and (9), and



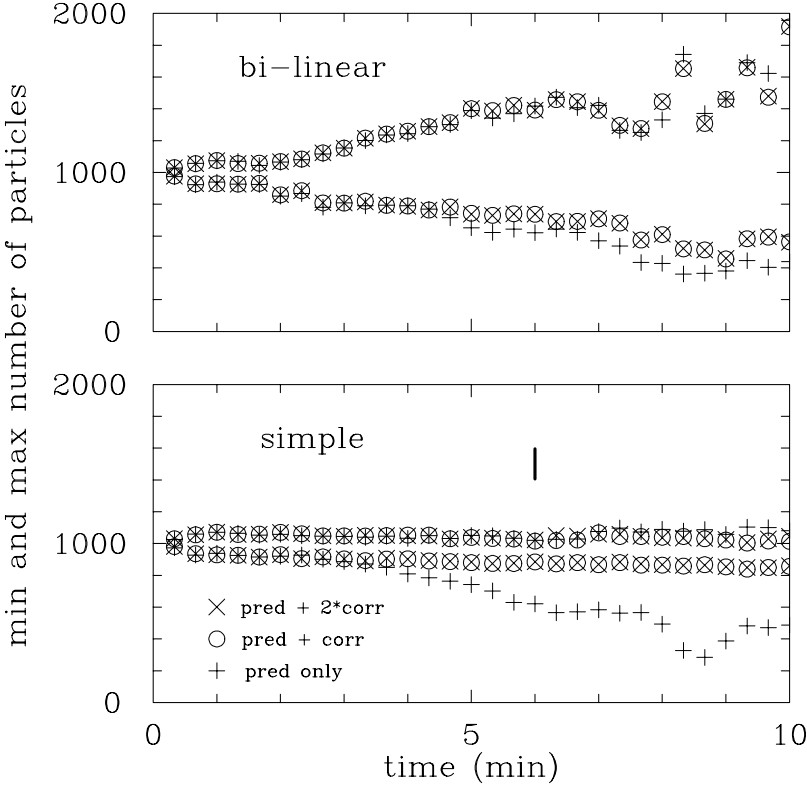

**Figure 4.** Evolution of the maximum and the minimum number of passive particles per grid cell in the bulk thermal simulations applying (upper panel) the bilinear advection scheme and (lower panel) the 'simple' scheme (Fig. 3). Both schemes apply either the predictor step or the predictor-corrector with either one or two corrective iterations. The vertical bar length in the lower panels corresponds to six standard deviations of the expected number of particles per grid cell. See text for details.

the predictor-corrector scheme with additional iteration of (9). The initial number of passive particles is 1,000 per grid cell. The standard deviation of the number of particles per grid cell after advection should be close to the square root of the initial number, that is, close to 30 (around 3%) for the simulations shown in Fig. 4. It follows that the difference between the maximum and minimum number of passive super-droplets per grid cell should be not significantly larger than a few standard deviations.

5   As the upper panel of Fig. 4 shows, the bi-linear interpolation scheme leads to a significantly larger difference starting around minute 2 of the simulation. This is clearly unphysical as argued above. In contrast, the simple scheme with a single corrective iteration provides physically-consistent results, that is, the difference between the maximum and minimum is several times the standard deviation of the initial particle number per grid cell and such a difference is maintained throughout the 10-min long simulation. Only an insignificant improvement is simulated with the additional iteration of the corrective step (9). Finally, a

10   slight reduction of the mean concentration (located somewhere between the maximum and minimum symbols) is apparent in the bottom panel of Fig. 4. This is because of the reduction of the droplet concentration due to the decrease of the air density with height (i.e., the term on the right hand side of Eq. 9).



This simple example, together with similar simulations using different numbers of passive particles not shown here as well as results of the super-droplet approach available at the University of Warsaw (Arabas et.al. 2015) all suggest that the simple scheme (10) (and its extension into 3D framework) should be used in the Lagrangian microphysics. Hence, such a scheme is used in all super-droplet simulations presented in this paper.

## 2.5   Coupling thermodynamic Eulerian and Lagrangian fields

The overall strategy for the time integration of the coupled Eularian and Lagrangian components of the model thermodynamics is to advance the temperature and moisture fields using (6) first, then to transport Lagrangian super-droplets using (7) and (8), and finally to calculate growth/evaporation of cloud droplets according to (4), with the growth/evaporation providing temperature and moisture tendencies calculated from (3) in each grid cell. These tendencies are applied in the next model time

step. Growth/evaporation of individual super-droplets requires knowledge of the spersaturation that needs to be calculated from updated temperature and water vapor fields. The flow-resolved supersaturation field can be supplemented with the subgrid-scale fluctuations (for instance, as in Grabowski and Abade, 2017). By the same token, the resolved flow used to transport super-droplets through the predictor-corrector scheme can be supplemented with the subgrid-scale velocity fluctuations estimated from the predicted subgrid-scale turbulent kinetic energy. These additions are not included in the initial formulation and testing

of the Twomey super-droplets discussed in this paper, but will form an important component of the model application in the future.

There are two issues that need to be considered for the coupling between Eulerian and Lagrangian model components. The first one concerns spurious supersaturation fluctuations near cloud edges (see Grabowski and Morrison, 2008, and references therein). This problem is particularly serious when the Twomey activation is used as illustrated later in the paper because of the

direct link between local supersaturation and the concentration of activated cloud droplets. Specifically, numerical overshoots of the supersaturation lead to an immediate activation of new cloud droplets. In contrast, when deliquescence and droplet activation is explicitly considered in the traditional super-droplet method, these transient overshoots may have smaller impact on the droplet activation. This is one of the conclusions of the Hoffmann (2016) study, also confirmed by simulations discussed in this paper. Grabowski and Morrison (2008) developed a relatively simple method to cope with this problem for the case of

a double-moment Eulerian microphysics scheme and suggested how it can be extended to the bin microphysics. We apply the Grabowski and Morrison (2008) methodology to the super-droplet simulations as discussed below.

The second issue concerns the interpolation of the thermodynamic fields to the super-droplet position. Shima et al. (2009) (see section 5.1.2) Riechelmann et al. (2012) (section 2.2.3) and Andrejczuk (personal communication 2017) interpolate the potential temperature and water vapor mixing ratio and then derive the local supersaturation. Such an approach is not appropriate

due to the nonlinear relationship between the supersaturation and the potential temperature. Interpolating the supersaturation would be more appropriate. However, supersaturation interpolation brings conceptual issues similar to those concerning super-droplet transport: if a single super-droplet represents a large ensemble of real cloud droplets, should growth of the ensemble be represented using the grid-averaged conditions? Moreover, one-dimensional tests with stationary cloud-environment interface show that the supersaturation interpolation results in a gradual erosion of the cloud edge. This is because supersaturation





interpolation between a cloudy grid cell near the cloud edge and a sub-saturated cell outside the cloud results in sub-saturated conditions for super-droplets located near the cell boundary leading to their evaporation. In contrast, applying the mean super-saturation maintains the steady conditions near the motionless cloud-environment interface. Moreover, applying the Grabowski and Morrison (2008) methodology to cope with the spurious cloud-edge supersaturation discussed below becomes cumber-
some (if not impossible) when the supersaturation interpolation to the super-droplet position is used. Overall, our tests similar to those discussed in the next section suggest that the impact of supersaturation interpolation in a rising thermal simulations is small and thus we decided to proceed with the simpler and computationally more efficient method of applying the grid-cell supersaturation to growth/evaporation of all super-droplets within a given grid cell.

### 2.6  Avoiding spurious cloud-edge supersaturations

The key aspect of the Grabowski and Morrison (2008) (GM08 hereinafter) method is to rely on the prediction of the absolute supersaturation (the difference between the water vapor mixing ratio and its saturated value) and to locally adjust the water vapor, cloud water, and temperature to maintain the predicted absolute supersaturation. This is in the spirit of Grabowski (1989) who used the temperature and supersaturation as main model variables and diagnosed the water vapor mixing ratio. Such a method results in a physically consistent supersaturation field but does not conserve water. GM08 circumvent this problem and
apply the approach to the Eulerian double-moment cloud microphysics (i.e., predicting number and mass mixing ratios of the cloud water field). They also suggest how this approach can be used in the bin scheme (see section 4 therein). Here we explain how this method is used with Twomey super-droplets.

The crux of the method is to calculate the amount of cloud water $\epsilon$ that needs to condense or evaporate to ensure that the predicted potential temperature and water vapor mixing ratio fields give the absolute supersaturation that agrees with the
predicted one. Thus, in addition to the prediction of the potential temperature and water vapor mixing ratio, the scheme predicts the evolution of the absolute supersaturation (see Eq. A8 in Grabowski and Morrison, 2008, and Eq. 4 in GM08). Once the amount of cloud water involved in the adjustment is calculated as in (7) of GM08, one needs to decide how that amount is distributed among super-droplets present within a given grid cell. Following GM08, the amount of cloud water $\epsilon$ that needs to be distributed among N super-droplets from a given cell is calculated as

$$\epsilon = \sum_{i=1}^{N} \epsilon_i \,, \tag{11}$$

$$\epsilon_i = \frac{\epsilon}{\beta \tau_i} \,, \quad \text{where} \quad \beta = \sum_{k=1}^{N} \frac{1}{\tau_k} \,, \tag{12}$$

where $\tau_i$ is the phase relaxation time scale for the ith super-droplet (cf. A5 in Grabowski and Morrison, 2008). Knowing $\epsilon_i$, the radius of each super-droplet within a given grid cell is subsequently modified keeping the multiplicity parameter the same.



## 3 Example of application: 2D moist thermal simulations

The scheme described above has been merged with the EULAG model (e.g., Prusa et al., 2008, www2.mmm.ucar.edu/eulag/) and its simplified version referred to as babyEULAG (Grabowski, 2014, 2015). Here we present results from the babyEULAG model as it is simpler and thus more convenient for the scheme testing and improvement. The University of Warsaw Lagrangian

Cloud Model (UWLCM) briefly described in the next section is used in the comparison. The Lagrangian approach applied in the UWLCM is referred to as the traditional super-droplet method in the discussion below. Both the babyEULAG model and the UWLCM apply the implicit large eddy simulation approach, that is, without modeling of the unresolved subgrid-scale transport (see references to other studies applying this method in  Grabowski, 2014; Pedersen et al., 2016).

### 3.1 The University of Warsaw Lagrangian Cloud Model, UWLCM

The UWLCM is an open-source software for 2D/3D modelling of clouds with super-droplet or bulk microphysics. Advection of the Eulerian fields is done using the libmpdata++ (Jaruga et al., 2015) implementation of the MPDATA algorithm (Smolarkiewicz and Margolin, 1998). Cloud microphysics is modelled using the libcloudph++ library (Arabas et al. 2015). Coupling between Eulerian and Lagrangian model components is done in the same way as in the Twomey model. Potential temperature and water vapor mixing ratio are not interpolated to the position of a super-droplet, but the same value is used for all

droplets within a cell. The procedure for limiting spurious supersaturation, which was described in Sec. 2.6, is not used. The super-droplets are advected with a predictor-corrector method with velocities interpolated to super-droplet position using the "simple" scheme defined in Sec. 2.4. In the MPDATA algorithm used in simulations presented in this paper, variable-sign fields were handled using the "abs" option of the libmpdata++ library (see Secs. 3.1.5 and 3.4.1 in Jaruga et al., 2015). Advection of Eulerian fields and of super-droplets is done with a $\Delta t = 1s$ timestep. Water condensation is done using 10 sub-steps per

timestep of advection, resulting in a $\Delta t = 0.1s$ timestep for condensation. The sub-steps are implemented in a per-particle manner, that is, each super-droplet remembers its own values of Eulerian fields from the previous advection timestep.

There are some differences in the super-droplets models used in UWLCM and in the Twomey model presented in this paper. In UWLCM, a more detailed equation for condensational growth is used (see Sec. 5.1.3 in Arabas et al., 2015). It utilizes the $\kappa$- Köhler parametrization of aerosol hygroscopicity. We assume $\kappa = 1.28$ for the sea salt aerosol used in this paper (see Tab. 1

in Petters and Kreidenweis, 2007). Another difference is that super-droplet multiplicity in UWLCM is the number (in contrast to the number mixing ratio in the Twomey model) of real droplets a given super-droplet represent. Moreover, super-droplets can have different multiplicities and the number of super-droplets is prescribed. The super-droplet initialisation scheme is the same as in Dziekan and Pawlowska (2017) (the "constant SD" type of simulation described in Sec. 2 therein).

### 3.2 Setup of moist thermal simulations

Rising moist thermal simulations follow Grabowski and Clark (1991) and Grabowski and Clark (1993) with small modifications. The environmental profiles are taken as constant stability $\mathrm{d}\ln\Theta_v/\mathrm{d}z = 1.3 \times 10^{-5}$ m$^{-1}$ for the temperature ($\Theta_v$ is the virtual potential temperature; the potential temperature based stability was used in Grabowski and Clark) and constant relative

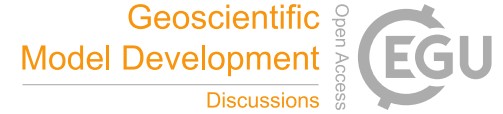

humidity of 20%. Surface temperature and pressure are taken as 283 K and 850 hPa. Note that $\Theta_v$-based stability profile requires iterative procedure when moving upwards from the surface because the temperature, moisture, and pressure (the latter resulting from the hydrostatic balance) have all to be adjusted to give stability and relative humidity profiles exactly as specified above. The circular moisture perturbation is introduced in the middle of the 3.6 km horizontal domain, with the center located

at the 800 m height. The vertical extent of the domain is 2.4 km. The air inside the 250 m perturbation radius (200 m was used in Grabowski and Clark) is assumed saturated, and the relative humidity decreases to the environmental value as cosine squared over the 100 m radial distance. Uniform horizontal and vertical grid length of 20 m is used. A 1-sec time step is used in simulations applying the babyEULAG model.

### 3.3    Comparison between UWLCM and the Twomey super-droplets

When comparing results from the two models, one needs to keep in mind that microphysical schemes differ in some additional details. In particular, the UWLCM applies the $\kappa$- Köhler parametrization (Petters and Kreidenweis, 2007) to prescribe CCN activation characteristics whereas the Twomey scheme applies the N-S relationship derived from activation calculations applying CCN chemical composition information (i.e., as in Grabowski et al., 2011). Our tests with the adiabatic parcel model applying either the $\kappa$- Köhler parametrization used in UWLCM or the approach based on the CCN chemical composition show

that a couple percent difference between droplet concentration predicted by the two methods for the same supersaturation is not unusual. Moreover, different droplet growth equations are used in the two schemes, although this factor does not affect the favorable comparison presented in Grabowski et al. (2011).

Figures 5 and 6 show spatial distributions of the water vapor and cloud water mixing ratios for the two simulations, that is, using either the Twomey super-droplets with the babyEULAG model (Fig. 5) or the traditional super-droplets with UWLCM

model (Fig. 6). Both simulations apply similar number of super-droplets per grid cell. It is impossible to match the numbers exactly, as the number is specified directly in UWLCM and indirectly through the number of $S_{\max}$ divisions in the Twomey approach. Overall, the transition of the initial circular perturbation to a cloudy rising vortex pair proceeds similarly in the two models. The most obvious difference comes from the development of instabilities near the thermal top. These instabilities are forced by fluctuations of thermodynamic fields (and thus cloud buoyancy) that result from a finite number of super-droplets

in each grid cell. As discussed in Grabowski and Clark (1991, 1993), these cloud-environment instabilities represent a combination of Rayleigh-Taylor and Kelvin-Helmholtz instabilities occurring in a complex geometry near the thermal leading edge. The spatial scale of the instability depends on the depth of the shear that develops near the cloud-environment interface as the thermal pushes upwards (see section 4b in Grabowski and Clark, 1991). The specific realization of the instability pattern changes with the number of super-droplet used in the simulation, and with the selection of random numbers applied during

positioning super-droplet on the Eulerian grid during activation. It follows the direct comparison between the simulations is possible only before the development of the instabilities, say, up to the 6th minute of the simulation (i.e., middle panels in Figs. 5 and 6 ).

A more detailed comparison between the two simulations is facilitated applying two different statistical measures. The first one involves conditional sampling of various fields across the thermal including points with the cloud water mixing ratio





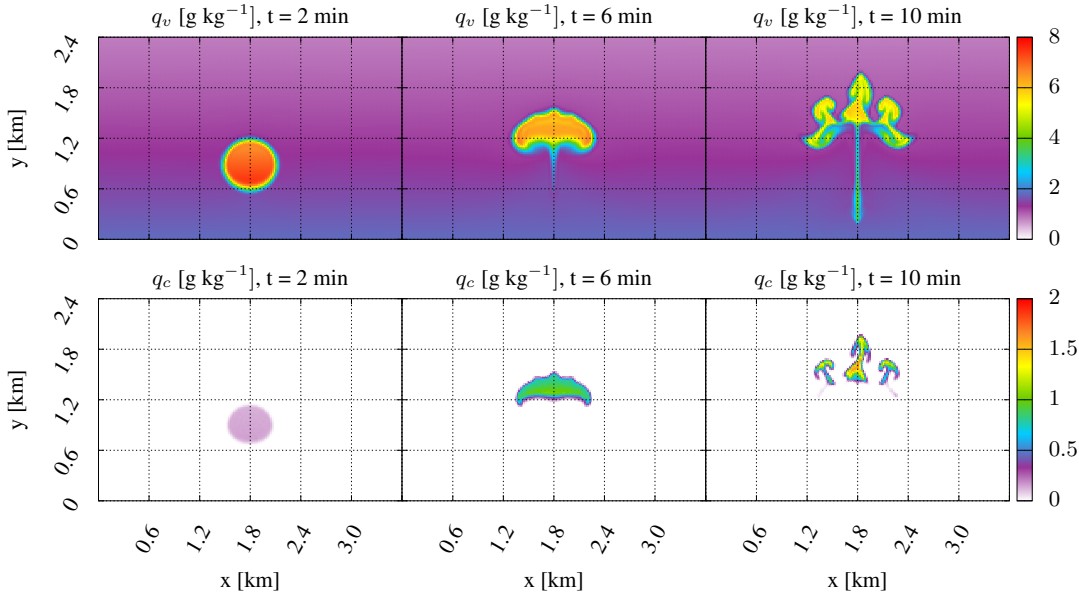

**Figure 5.** Distribution of (upper panels) the water vapor and (lower panels) the cloud water mixing ratios at 2, 6 and 10 min for the Twomey super-droplet scheme in the rising thermal simulation.

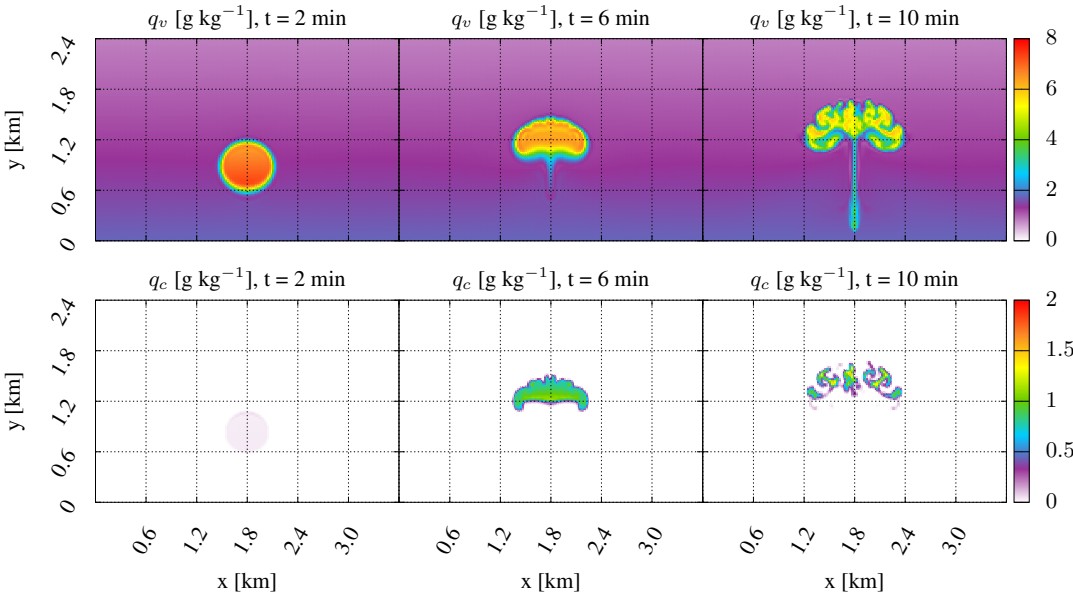

**Figure 6.** As Fig. 5, but for the UWLCM model.

exceeding a threshold of 0.1 g/kg. A smaller threshold allows incorporation of a more significant fraction of points from the thermal edge that are affected by the Eulerian model numerics. The statistics include the mean values of conditionally sampled





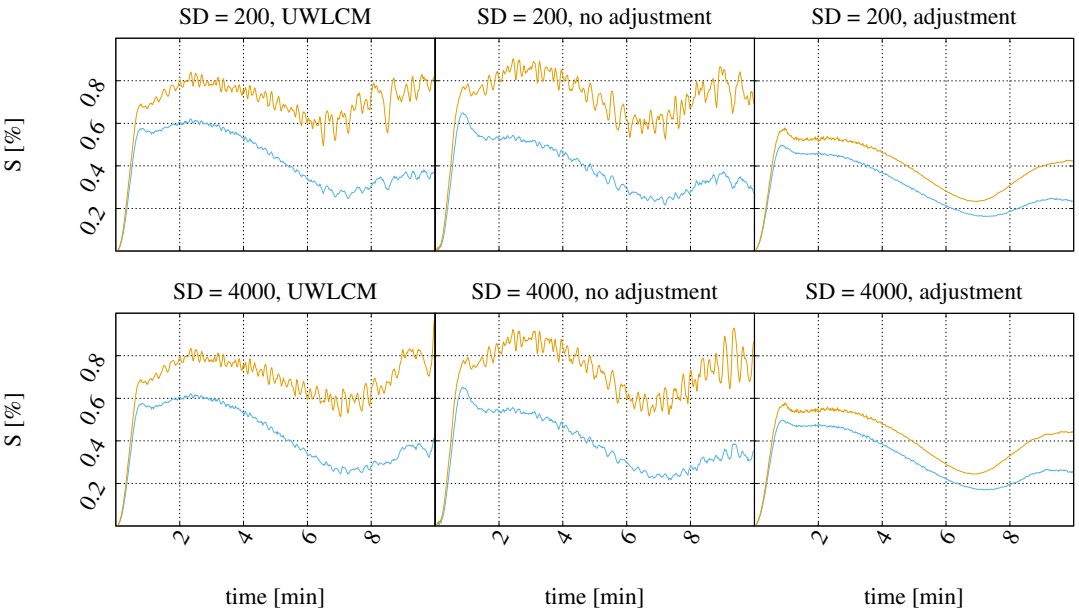

**Figure 7.** Comparison of the mean supersaturation averaged over the cloudy points for UWLCM (left panels) and the Twomey super-droplet scheme without (middle panels) and with (right panels) the adjustment to avoid unphysical cloud-edge supeersaturation fluctuations. The blue and brown lines represent the mean and the mean plus the standard deviation of the spatial distribution. Upper and lower panels are for simulations with 200 and 4000 super-droplets (UWLCM) or number of divisions (Twomey scheme). Data is plotted every time step of the fluid flow model.

fields and the standard deviations of the spatial variability of a given field across the thermal. The second measure is the time evolution of various quantities at the center of mass of the cloud water field, that is, at the height of $z_{cm} = \int z q_c \, ds / \int q_c \, ds$ (where $q_c$ depicts the cloud water mixing ratio and the integral is over the entire computational domain) and a similar expression for the horizontal position $x_{cm}$. Simulations with different number of super-droplets were considered, although a direct match

of the super-droplet number per grid cell between UWLCM and the Twomey is impossible as explained previously. In Twomey scheme simulations described here, the number of $S_{max}$ divisions considered was 50, 200, 1000 and 4000. The corresponding number of super-droplets was around 40 with the standard deviation of around 10 for 50 divisions; $\sim$150 and $\sim$30 for 200 divisions, $\sim$700 and $\sim$110 for 1000 divisions, and $\sim$2700 and $\sim$400 for 4000 $S_{max}$ divisions. As explained previously, the number of super-droplets per grid cell affects the amplitude of oscillations due to the transport of super-droplets across the

Eulerian grid.

    Figure 7 compares evolutions of the supersaturation field conditionally sampled over the rising thermal for UWLCM with 200 and 4000 super-droplets per grid cell and the Twomey approach with 200 and 4000 $S_{max}$ divisions. The Twomey results are with either applying or excluding (marked adjust and noadjust in the figure) temperature and moisture adjustment as described in section 2.6. The figure shows that application of the adjustment scheme is critical for maintaining physically-consistent su-

persaturation field as the supersaturations are significantly higher without the adjustment. The physical consistency is measured



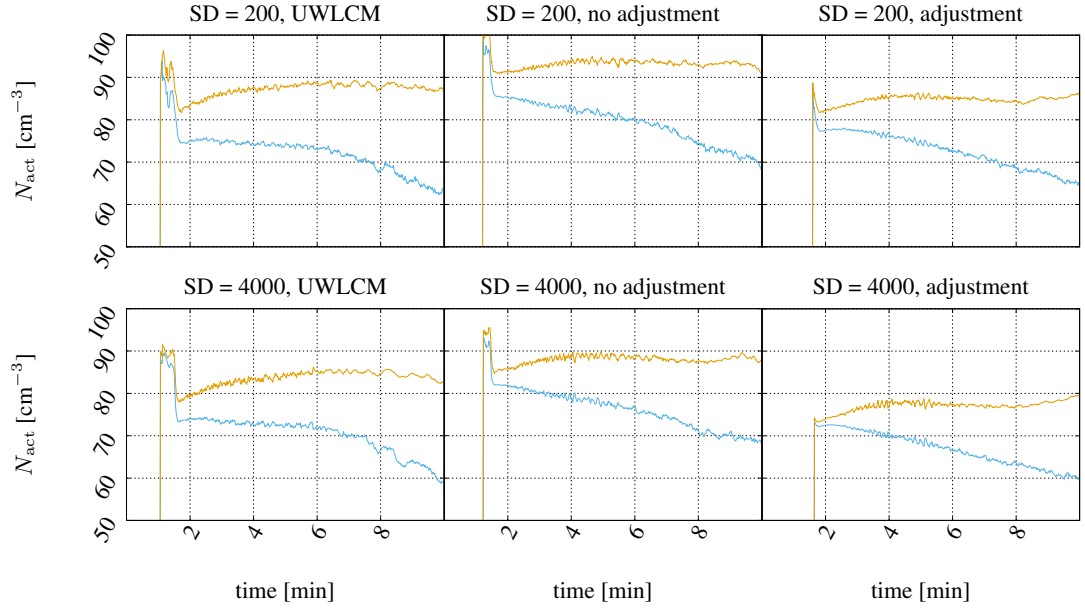

**Figure 8.** As in Fig. 7, but for the droplet concentration $N_{act}$.

by comparing the supersaturation predicted locally with the quasi-equilibrium supersaturation, that is, the supersaturation resulting from a balance between production due to the updraft and removal due to condensation (Squires, 1952; Politovich and Cooper, 1988). Except for the initial first minute when droplets are small, the supersaturation predicted by the model agrees well with the quasi-equilibrium supersaturation (not shown). An important point is that simulations with 200 divisions (upper panels) differ little from simulations applying 4000 divisions (lower panels) until different flow realizations in the final few minutes cause the divergence. The agreement suggests that about a hundred super-droplet per grid cell is sufficient to obtain statistics that change little with further increase of the super-droplet number. The reduction of both the standard deviation and the amplitude of the fluctuations when the adjustment scheme is applied is also apparent. The corresponding results from UWLCM simulations show that the mean supersaturation evolution is similar to those for the Twomey simulations without adjustment as one might expect.

Figure 8 shows statistics of the droplet concentration in the format similar to Fig. 7. As expected, the mean concentration is higher when adjustment is not used in the Twomey approach. The mean concentration slowly decreases in time because of the air expansion due to rise of the thermal. For Twomey simulations with 200 $S_{max}$ divisions, the mean concentration at minute 2 is around $78\,\mathrm{cm}^{-3}$ and the standard deviation of the spatial distribution is around $5\,\mathrm{cm}^{-3}$ in simulation with the adjustment versus $85\,\mathrm{cm}^{-3}$ and higher standard deviation without the adjustment. The mean concentration at 2 minutes decreases to around $73\,\mathrm{cm}^{-3}$ for the 4000 division simulations. For the UWLCM, the mean concentration at minute 2 is around $75\,\mathrm{cm}^{-3}$ and the standard deviation of the spatial distribution is around $9\,\mathrm{cm}^{-3}$. The similarity of the mean concentration between the Twomey super-droplets with adjustment and UWLCM documents the limited impact of the cloud-edge supersaturation



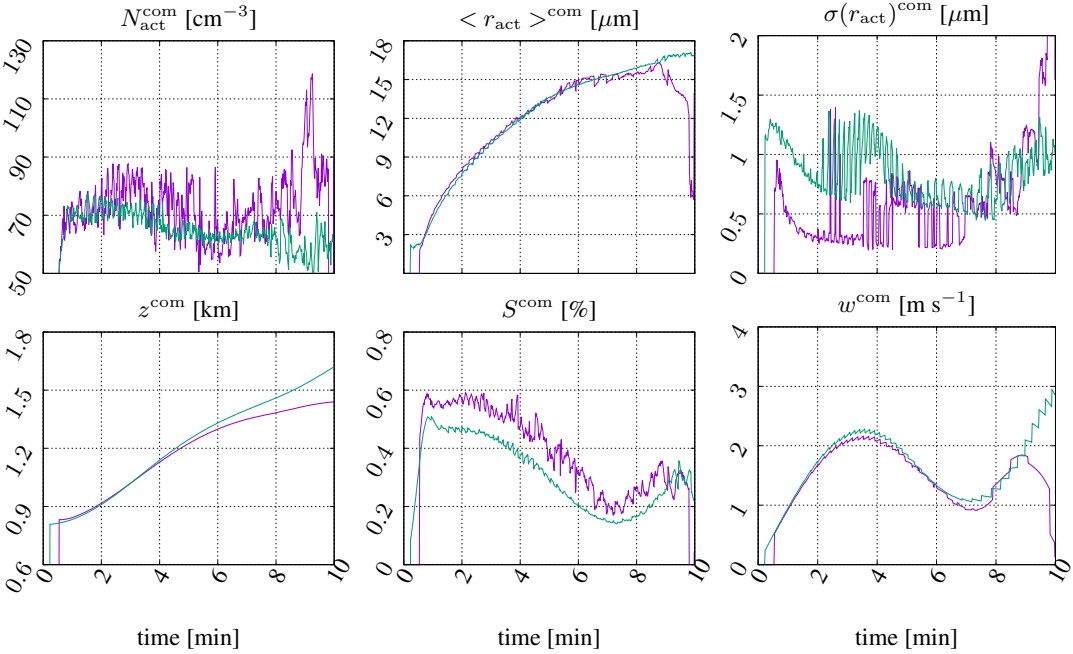

**Figure 9.** Evolution of various parameters at the center of mass of the cloud water in the rising thermal simulations applying 200 super-droplets (UWLCM; purple lines) or 200 divisions (Twomey scheme with adjustment; green lines). The panels show: (upper left) droplet concentration; (upper middle) droplet mean radius; (upper right) spectral width of the droplet size distribution; (lower left) height of the center of mass; (lower middle) supersaturation; and (lower right) vertical velocity.

fluctuations when details of the CCN activation are resolved in the original super-droplet approach. Larger standard deviation of the spatial distribution is likely because of the variable multiplicity attribute among original super-droplets in the UWLCM model.

Figures 9 and 10 compare various statistics between the Twomey scheme and UWLCM at the center of mass of the cloud

5   water field with 200 and 4000 super-droplets (UWLCM) or $S_{\mathrm{max}}$ divisions (Twomey scheme), 200 in Fig. 9 and 4000 in Fig. 10. The data are plotted every model time step. Microphysical properties such as the droplet concentration, mean radius and the spectral width show oscillations that are reduced with the increased number of super-droplets per grid cell. This is a consequence of the impact of droplet transport from one grid-cell to another: the larger the number of super-droplets the smaller the oscillations. The amplitude of the oscillations can be estimated by comparing the original evolution (i.e., the one

10   shown in Figs. 9 and 10) and the evolution sufficiently smoothed in time so the oscillations are removed. In the case of the droplet concentration for the Twomey simulations, the amplitude decreases from  6.0,  2.0,  1.1, and $0.6 \ \mathrm{cm}^{-3}$ for the number of divisions increasing from 50, 200, 1000 to 4000. This is roughly the expected scaling, that is, along the square root of the number of super-droplets that increase from around 40 to 2700 for the number of divisions increasing from 50 to 4000. As mentioned before, the direct comparison between various simulations is only possible up to about 6th minute as different flow





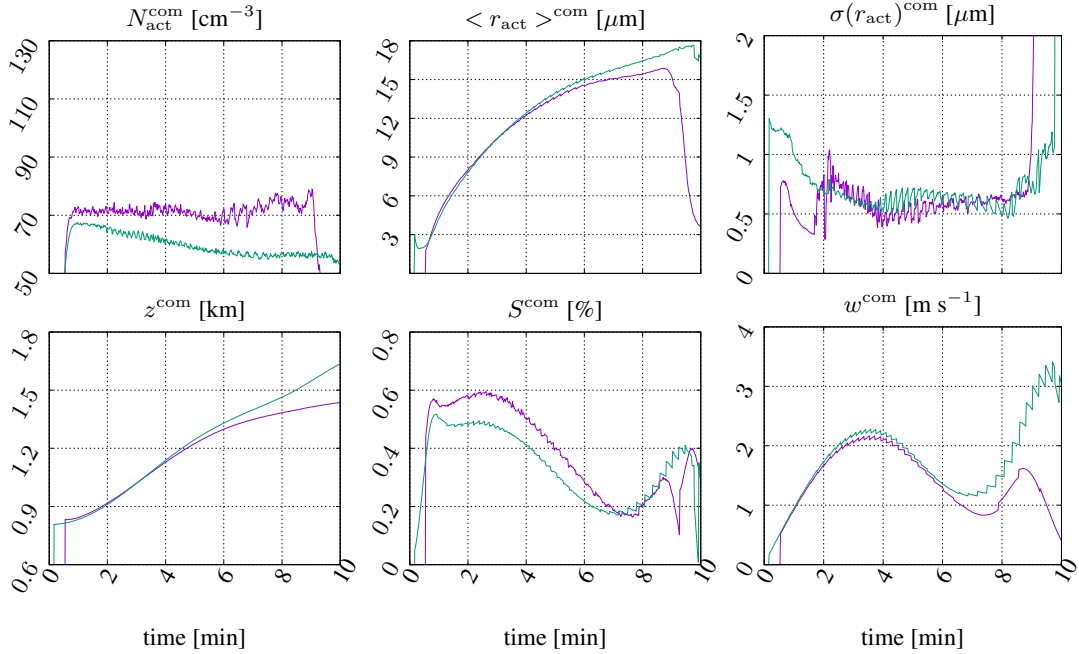

**Figure 10.** As Fig. 9, but for 4000 super-droplets (UWLCM) or 4000 divisions (Twomey scheme).

evolutions make results impossible to compare at later times. Except for the oscillation amplitude, the results for different number of $S_{max}$ divisions compare well for the Twomey super-droplets.

The differences between Figs. 9 and 10 are consistent with the differences between conditionally-averaged statistics. For instance, droplet concentrations fluctuate between 60 and 80 cm$^{-3}$ for Twomey super-droplets with adjustment and 200 $S_{max}$

divisions and UWLCM with 200 super-droplets per grid cell. The evolution of the center of mass height is very similar in Twomey and UWLCM simulations. The mean radius is close to 15 $\mu$m at minute 6 for both the Twomey and UWLCM. Droplet concentration fluctuations are larger in UWLCM arguably because of the way CCN is sampled when super-droplets are created, that is, with different multiplicity parameter that increases the oscillation amplitude. The evolution of the vertical velocity at the cloud water center of mass increases in both simulations up to about 3.5 min and decreases thereafter (the

evolutions after minute 6 cannot be compared due to different flow realizations as already explained). The vertical velocity maxima around 3.5 min are similar.

As a final element of the comparison, we show in Fig. 11 evolutions of the maximum supersaturation in the computational domain for simulations with the Twomey super-droplets with and without the adjustment to limit un-physical cloud-edge supersaturations and for the UWLCM. The simulations apply similar number of super-droplets per grid cell (200 divisions in

the Twomey super-droplet simulations and 200 samples of the CCN distribution in the UWLCM simulation). The maximum supersaturations occur at different spatial locations near cloud edge as the cloud-environment interface moves across the Eulerian grid. This is why large fluctuations in the UWLCM simulation impact the mean droplet concentration to a smaller degree



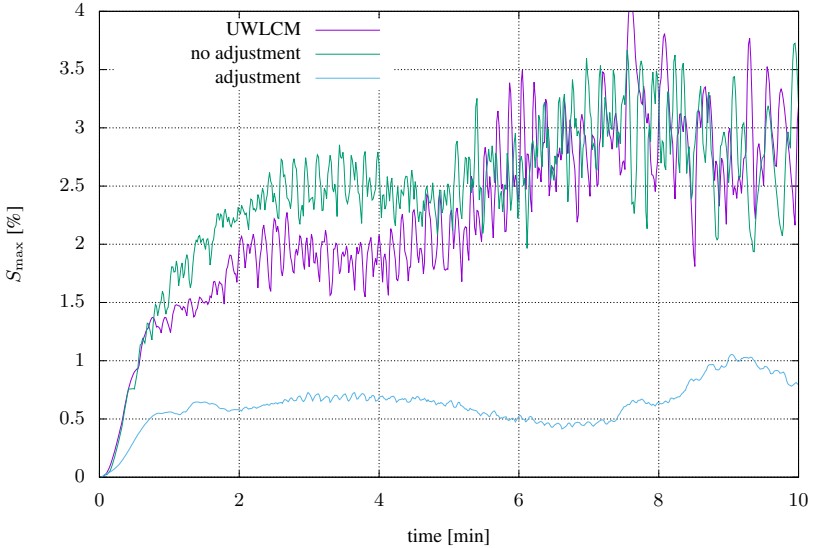

**Figure 11.** Evolution of the maximum supersaturation in the domain for UWLCM simulations (purple line) and Twomey scheme with (blue line) and without (green line) adjustment for unphysical supersaturation fluctuations. Data every model time step with 200 super-droplets/divisions.

than in the Twomey approach. In a nutshell, CCN has no time to respond to these fluctuations when deliquescence is explicitly calculated by the model. In contrast, Twomey activation immediately adds new droplets when supersaturation fluctuations take place. These additional droplets can evaporate in subsequent times steps, but some survive and lead to the increased mean droplet concentration as documented in Fig. 8. It follows that the adjustment is the key element of the Twomey super-droplets,

5 but is less significant for the traditional super-droplet approach, see Hoffmann (2016).

In summary, we believe that simple tests presented in this section document efficacy of the super-droplet approach with the Twomey activation. Unfortunately, we cannot provide a direct comparison of the computational effort between the two approaches because the two models run on different computer systems. However, since the cloud covers about 2.5% of the 2D computational domain, the Twomey scheme requires roughly 40 times less computational effort for simulations presented here.

10 However, for a hypothetical 3D simulation with a domain extending 3.6 km in the second horizontal direction, the volume of the initial spherical bubble with the same radius would only constiue about 0.1% of the computational domain volume. Thus, the computational effort in similar 3D simulations would be about three orders of magnitude larger in UWLCM than in the babyEULAG with Twomey super-droplets. UWLCM makes up a lot of this difference applying modern software engineering techniques including parallel processing and application of graphics processing units, see Arabas et al. (2015).



## 4  Conclusions

This paper discusses technical details of a novel Lagrangian condensation scheme to model non-precipitating warm (ice-free) clouds. The idea is to use Lagrangian point particles ("super-droplets" following the nomenclature introduced by Shima et al. 2009), rather than continuous medium variables such as number or mass mixing ratios, to represent condensed cloud water. Previous studies applying such methodology (e.g., Andrejczuk et al., 2008, 2010; Shima et al., 2009; Riechelmann et al., 2012; Arabas et al., 2015; Hoffmann et al., 2015) demonstrate significant advantages of the super-droplet method, such as reduced numerical diffusion, formulation of the governing equations from the first principles, and straightforward application of suitable statistical techniques to represent unresolved subgrid-scale variability (e.g., Grabowski and Abade, 2017). However, in previous applications the Lagrangian microphysics, the super-droplets outside clouds represent un-activated CCN that become activated upon entering a cloud and can further grow through diffusional and collisional processes. Thus, the super-droplets fill the entire computational domain and need to be transported even if they exist far away from a cloud and do not affect cloud processes. The original methodology allows studying in detail not only effects of CCN on cloud microphysics and dynamics, but also CCN processing by a cloud. When applying the super-droplet method to problems where CCN processing is of secondary importance (e.g., the impact of entrainment on the spectrum of cloud droplets), a simpler and computationally more efficient approach can be used. The idea is to create super-droplets only when CCN is activated and to remove them when a complete evaporation (i.e., CCN de-activation) takes place. Thus, no super-droplet exists outside a cloud and a significantly smaller number of super-droplets need to be followed in space and time when compared to the traditional super-droplet scheme with the same number of super-droplets per grid cell. The new super-droplet approach is possible by applying the Twomey activation method where the local supersaturation dictates the concentration of cloud droplets (and thus the number of the super-droplets) that need to be present inside a cloudy volume. Twomey activation excludes details of the CCN deliquescence and activation, and super-droplets simply disappear when a complete evaporation of cloud droplets occurs. Twomey activation is often used in Eulerian bulk (e.g., Morrison and Grabowski, 2007, 2008) and bin microphysics schemes (e.g., Grabowski et al., 2011; Wyszogrodzki et al., 2011). Moreover, simulation of the CCN deliquescence requires short time steps (especially for small CCN) and avoiding it with the Twomey activation provides additional computational advantage.

We apply the traditional Lagrangian super-droplet model, the University of Warsaw Lagrangian Cloud Model (UWLCM; Arabas et al., 2015; Jaruga et al., 2015) and compare results from UWLCM and the novel Twomey super-droplet method. The simulations apply an idealized setup of a moist thermal rising in a stratified environment (Grabowski and Clark, 1991, 1993). Overall, the comparison demonstrates the efficacy of the new approach as simulation results differ little between UWLCM and the new scheme. This is consistent with adiabatic parcel results discussed in Grabowski et al. (2011) that - away from the cloud base - show good agreement between cloud properties simulated applying a scheme with Twomey activation and a scheme where details of the CCN deliquescence are modeled explicitly. The results presented here show that avoiding spurious cloud-edge supersaturation fluctuations is essential with the Twomey activation. This is because these fluctuations immediately translate into unphysical droplet concentrations that affect subsequent evolution of cloud microphysical properties. In contrast,



these highly transient in space and time fluctuations seem to have small impact on simulations using the original super-droplet method, in agreement with results discussed in Hoffmann (2016).

As noted in Clark (1974), Morrison and Grabowski (2008) and Grabowski and Jarecka (2015) , modeling cloud base activation in the Eulerian cloud model requires high vertical resolution to resolve cloud base supersaturation maximum, say, of the

order of 10 m. The same is true for the Lagrangian super-droplets. In the case of lower vertical resolution (i.e., when the cloud base supersaturation maximum is poorly resolved), an activation parameterization can be used, for instance, linking the concentration of activated CCN to the strength of the updraft velocity (e.g., Abdul-Razzak et al., 1998; Abdul-Razzak and Ghan, 2000, among others). Such a parameterization can also be used with the methodology presented in this paper, for instance, in simulations of deep convection that only allow low vertical resolution. As deep convection requires incorporation of ice physics into

the Lagrangian methodology, the possibility of applying even simpler representation of super-droplet formation through the activation parameterization is appealing. Such a methodology will pave the way for applications of the Lagrangian methodology beyond high-spatial resolution large eddy simulation today to the cloud-resolving (convection-permitting) weather and climate simulation of the future. We plan to include such developments to the Twomey super-droplet scheme presented here, together with the inclusion of the collision/coalescence that will be the focus of the immediate scheme expansion. These developments

will be reported in forthcoming publications.

*Code availability.* A Fortran code was used to perform Twomey super-droplet simulations. The code is available from the first author upon request.

**Appendix A: Subgrid-scale divergence of the simple interpolation scheme.**

Fig. A1 shows a grid cell with a subgrid volume of dimensions $\Delta\alpha$ and $\Delta\gamma$ with velocities $u_1$, $u_2$, $w_1$ and $w_2$ on the volume

boundaries. Using the simple interpolation scheme (10) and symbols defined in Fig. A1, the horizontal velocities $u_1$ and $u_2$ are given by:

$$u_1 = (\alpha - \frac{\Delta\alpha}{2})u_{i+\frac{1}{2},k} + (1 - \alpha + \frac{\Delta\alpha}{2})u_{i-\frac{1}{2},k} , \tag{A1}$$

$$u_2 = (\alpha + \frac{\Delta\alpha}{2})u_{i+\frac{1}{2},k} + (1 - \alpha - \frac{\Delta\alpha}{2})u_{i-\frac{1}{2},k} , \tag{A2}$$

and the vertical velocities $w_1$ and $w_2$ are:

$$w_1 = (\gamma - \frac{\Delta\gamma}{2})w_{i,k+\frac{1}{2}} + (1 - \gamma + \frac{\Delta\gamma}{2})w_{i,k-\frac{1}{2}} , \tag{A3}$$

$$w_2 = (\gamma + \frac{\Delta\gamma}{2})w_{i,k+\frac{1}{2}} + (1 - \gamma - \frac{\Delta\gamma}{2})w_{i,k-\frac{1}{2}} . \tag{A4}$$



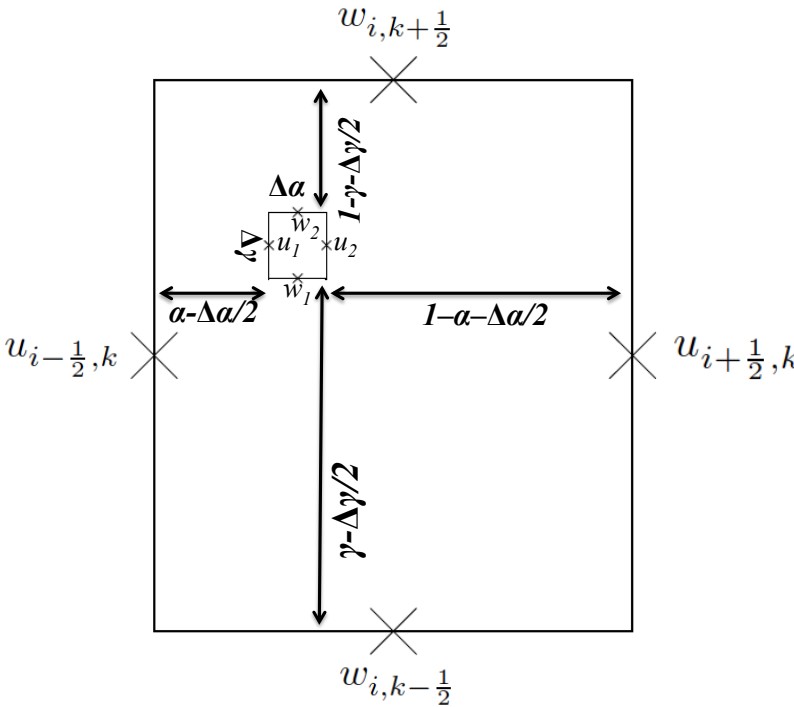

**Figure A1.** Interpolation of particle advecting velocities from the grid cell (large rectangle) into a subgrid-scale volume (small rectangle) applying the simple interpolation scheme (Fig. 3).

The divergence over the small volume (with dimensional extensions of $\Delta\alpha\Delta x$ and $\Delta\gamma\Delta z$) is then given by:

$$\frac{u_2 - u_1}{\Delta\alpha\Delta x} + \frac{w_2 - w_1}{\Delta\gamma\Delta z} = \frac{u_{i+\frac{1}{2},k} - u_{i-\frac{1}{2},k}}{\Delta x} + \frac{w_{i,k+\frac{1}{2}} - w_{i,k-\frac{1}{2}}}{\Delta z} = -\frac{w}{\rho}\frac{\partial\rho}{\partial z}. \tag{A5}$$

It follows that the divergence over the subgrid volume is exactly the same as over the grid cell volume. This is the key feature of the simple interpolation scheme (10) because it allows transport of super-droplets in a physically consistent manner as documented in the passive particle advection tests (see Fig. 4).

*Author contributions.* WWG developed the Twomey activation module and completed simulations using it. PD completed UWLCM simulations and created intercomparison figures. All authors contributed to the design of numerical simulations and were involved in creating the manuscript.

*Competing interests.* The authors declare that they have no conflict of interest.



*Acknowledgements.* This work was partially supported by the Polish National Science Center (NCN) "POLONEZ 1" Grant 2015/19/P/ST10/02596 and by the U.S. DOE ASR Grant DE-SC0016476. The POLONEZ 1 grant has received funding from the European Union's Horizon 2020 Research and Innovation Program under the Marie Sklodowska-Curie Grant Agreement 665778. WWG acknowledges discussions with Dr. Dorota Jarecka during the early development of Twomey super-droplet scheme. Helpful conversations and suggestions from Dr. Shin-Ichiro

5   Shima (U. of Hyogo, Kobe, Japan) and from Prof. Lian-Ping Wang (U. of Delaware, Newark, USA) are also acknowledged. UWLCM development has been possible through NCN grants 2010/01/N/ST10/01438, 2012/06/M/ST10/00434 and 2014/15/N/ST10/05143.



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
