# Peer review of "Lagrangian condensation microphysics with Twomey CCN activation"

_Geoscientific Model Development, 2017_

## Referee Comment (RC1) · Anonymous Referee #1 · 19 Sep 2017

**Review of "Lagrangian condensation microphysics with Twomey CCN activation" by Grabowski et al.**

The manuscript presents a new approach for the treatment of aerosol activation in Lagrangian Cloud Models (LCMs), a novel and promising approach for the simulation of cloud microphysics based on individually simulated super-droplets (SDs). The novelty (and advantage) of the new activation approach is that SDs are only introduced if the supersaturation exceeds a certain threshold. This is fundamentally different from previous activation approaches, in which SDs needed to be simulated even before activation. As pointed out in the manuscript, this new approach is not suited for simulating cloud-processing of aerosols. Applications in which cloud-processing of aerosols is of minor interest, however, benefit from reduced computing time as well as a smaller memory demand. Furthermore, the authors introduce further improvements necessary for the correct determination of supersaturations in LCMs: a velocity interpolation scheme which conserves the incompressibility of the flow as well as a technique to cope with spurious supersaturations. Since these latter refinements do not require the proposed aerosol activation scheme, they are a recommendable addition to all current LCMs.

All in all, this manuscript is well written, presents novel and useful methods, and is of interest to the entire LCM community. Accordingly, I recommend publishing this paper in Geoscientific Model Development. However, I would like the authors to address some minor comments, which will only increase the value of this already nice manuscript.

**Minor Comments**

- p. 2, l. 25: Please define "multiplicity". It might be understandable but there are also synonyms used in the literature (e.g., weighting factor).

- p. 4, Eq. (4): There is a "+" missing between "$r_i$" and "$r_0$".

- sec. 2.3: Although the Twomey activation approach is new to warm-cloud LCMs, there is already an analog in ice-cloud LCMs. Sölch and Kärcher (2010) describe how they introduce new SIPs (simulation ice particles – the ice-cloud equivalent to SDs) to the model domain based on an underlying nucleation scheme, which exhibits many similarities to Twomey activation. Additionally, Unterstrasser and Sölch (2014) describe how a stochastic representation of that nucleation scheme can improve the model's statistics. I think these publications should be mentioned and discussed in the manuscript.

- p. 6, ll. 11 – 12: Is the sentence "Without … past." true if entrainment/mixing is considered? The diluted number mixing ratio cannot reveal the previous maximum supersaturation. (Although the Twomey activation scheme will still be applicable.)

- p. 7, ll. 12 – 14: I agree with the sentence "This is … to another". However, the same multiplicity for all SDs might be disadvantageous for the initiation of collision and coalescence (see Unterstrasser et al., 2017).

- sec. 2.4: The suggested interpolation scheme should be used in all LCMs. However, there is one suggestion: Please add two plots to Fig. 4, which show the results for 100 SDs per grid cell, which is the typically applied SD concentration in current LCM simulations. This plot will be of great value to judge if there is a big impact of thoughtlessly applied tri-linear velocity interpolation in the published literature.

(Although I assume that there will be no impact visible due to the LCM's inherent fluctuations (now with a much higher standard deviation of around 10 %).)

- p. 10, l. 29: Why is the calculation limited to cloudy regions ($q_l > 0.01$ g kg$^{-1}$)? Shouldn't the results be independent of the region within the model domain?

- p. 14, ll. 19 – 21: Please give more details (or a reference) how the water condensation is split into 10 substeps. Based on the given text, I cannot imagine how this procedure is applied.

- sec. 3.2: Please add some details on the number of SDs initialized in each grid box or the maximum number of SDs per grid box created by the Twomey activation scheme. These details follow later (p. 17, l. 6) but I expected them to be in the setup section.

- Fig. 9/10: How do you define activated particles? Using the Twomey activation scheme, this is a straightforward task. But how do you proceed in the UWLCM?

- Fig. 9/10: Could you please comment a little more on the strong oscillations in the $\sigma(r_{act})^{com}$ plots? It seems that those time series jump between two solutions.

- p. 21, ll. 8 – 14: I agree that using the Twomey activation scheme will reduce the number of SDs in the model domain and, hence, computing time and memory demand. However, these considerations deserve some more thoughts. Models are usually parallelized using a 2D domain decomposition applied to the Eulerian fields but also the SDs. Accordingly, if there is a cloud in just in one subdomain, massive load imbalance will occur slowing down the whole computation. To benefit most of the new Twomey activation scheme, new parallelization strategies for the treatment of SDs need to be developed, e.g., a uniform distribution of SDs over all cores independent of their physical location in the model domain to avoid load imbalance issues.

**Technical Comments**

- p. 1, l. 18 (and several other places): The abbreviation SD has never been introduced.

- p. 2, l. 5 (and several other places): There is an unnecessary space between the bracketed citations and the following punctuation mark. Probably a LaTeX issue.

- p. 2, l. 30: I would cite Twomey's (1959) paper right here.

- p. 6, l. 5: A. Jaruga's PhD dissertation does not appear in the references section.

- p. 6, l. 17: The paper by Grabowsky and Abade (2017) has been cited several times in the unabbreviated form before the abbreviation GA17 is introduced.

**References**

Sölch, I., and Kärcher, B. (2010). A large-eddy model for cirrus clouds with explicit aerosol and ice microphysics and Lagrangian ice particle tracking, Q. J. Royal Meteorol. Soc., 136, 2074-2093.

Unterstrasser, S., and Sölch, I. (2014): Optimisation of the simulation particle number in a Lagrangian ice microphysical model, Geosci. Model Dev., 7, 695-709.

Unterstrasser, S., Hoffmann, F., and Lerch, M.: Collection/aggregation algorithms in Lagrangian cloud microphysical models: rigorous evaluation in box model simulations, Geosci. Model Dev., 10, 1521-1548.

---

## Short Comment (SC1) · 4 Oct 2017

Dear authors,

In my role as Executive editor of GMD, I would like to bring to your attention our Editorial version 1.1:

http://www.geosci-model-dev.net/8/3487/2015/gmd-8-3487-2015.html

This highlights some requirements of papers published in GMD, which is also available on the GMD website in the 'Manuscript Types' section:

http://www.geoscientific-model-development.net/submission/manuscript_types.html

In particular, please note that for your paper, the following requirements have not been

met in the Discussions paper:

- "The main paper must give the model name and version number (or other unique identifier) in the title."

- "All papers must include a section, at the end of the paper, entitled 'Code availability'. Here, either instructions for obtaining the code, or the reasons why the code is not available should be clearly stated. It is preferred for the code to be uploaded as a supplement or to be made available at a data repository with an associated DOI (digital object identifier) for the exact model version described in the paper. Alternatively, for established models, there may be an existing means of accessing the code through a particular system. In this case, there must exist a means of permanently accessing the precise model version described in the paper. In some cases, authors may prefer to put models on their own website, or to act as a point of contact for obtaining the code. Given the impermanence of websites and email addresses, this is not encouraged, and authors should consider improving the availability with a more permanent arrangement. After the paper is accepted the model archive should be updated to include a link to the GMD paper."

Thus please provide a name (acronym) and version number for your methodology in the title of the paper.

In the Code availability section the availability of all used model components is required. So please add information about EULAG and UWLCM. Additionally please note, that the exact versions of the model codes presented here (i.e. with the implementation of your methodology, must be available for review and should be archived permanently presumably via a permanent archive providing a DOI (e.g. Zenodo).

Yours,

Astrid Kerkweg

---

## Referee Comment (RC2) · S Shima (Referee) · 8 Oct 2017

**Lagrangian condensation microphysics with Twomey CCN activation**

Wojciech W. Grabowski[1,2], Piotr Dziekan[2], and Hanna Pawlowska[2]

[revised manuscript text omitted]

**T** Author: reviewer        Subject: Highlight        Date: 2017/10/08 21:16:19

4. Typo

d/dt
* * *
**T** Author: reviewer        Subject: Highlight        Date: 2017/10/08 21:16:24

5. Typo

\rho_vs
* * *
**T** Author: reviewer        Subject: Highlight        Date: 2017/10/08 21:16:36

6. Typo

r_i + r_0
* * *
**T** Author: reviewer        Subject: Highlight        Date: 2017/10/08 21:17:08

7. Minor request

Not the mass diffusivity Dv, but the thermal diffusivity \alpha should be used here. \alpha is about 15% smaller than Dv. See, e.g., table 2 of Montgomery (1947).
Please update eq (4) or justify the use of Dv here.
* * *
**T** Author: reviewer        Subject: Highlight        Date: 2017/10/08 21:17:20

8. Comment

As far as the collision/coalescence is not considered, I believe the interpretation of the super-droplet positions is clear: they are weighted random samples of the number density distribution of real-droplets.

Let $N_i$ be the number of RDs that the $i$-th SD is representing. Then, we can calculate the number density of RDs $n(\boldsymbol{x}, t)$ as follows.

$$n(\boldsymbol{x}, t) = \left\langle \sum_i N_i \delta^3(\boldsymbol{x} - \boldsymbol{x}_i(t)) \right\rangle, \tag{S1}$$

where the angle bracket denotes average over an ensemble of SD simulations generated by different random number sequences.

Let us consider a simple example. Assume that all the SDs have the same radius and multiplicity $N_i$. Also assume that all of them are distributed uniformly in a grid box at a time $t$. If 20% of the SDs go away from the grid box at the next time step $t + \Delta t$, this represents that 20% of the RDs go away from the grid. However, this is a stochastic event and suffers from random fluctuation. To eliminate this uncertainty, we need to calculate the average over an SD simulation ensemble.

[Figure]

interpretation is that a super-droplet represents all droplets located at a given moment in a particular grid cell. However, such an interpretation leads to a conceptual inconsistency because physically moving the super-droplet from one grid cell to another does not mean that all droplets that the particular super-droplet represents do the same. It is not clear if a physically consistent and computational efficient methodology can be developed to avoid this conceptual problem.

As in Andrejczuk et al. (2008, 2010); Shima et al. (2009) and Riechelmann et al. (2012) among others, the list of attributes for each super-droplet includes the position $\boldsymbol{x}_i$, radius $r_i$, and multiplicity. The latter depicts the number of particles represented by a single super-droplet. In the numerical implementation, the multiplicity is assumed to be the number mixing ratio, that is, $N_i/\rho_a$, similarly as in Eulerian bin microphysics schemes. Other attributes can be added if needed, for instance, the local supersaturation perturbation (on top of the grid-scale supersaturation predicted by the flow model) that can affect super-droplet growth in (4) as in Grabowski and Abade (2017) or the subgrid-scale velocity perturbation that can affect the motion of the super-droplet in (5).

**2.2 Numerical implementation**

As will be discussed in section 3, the novel super-droplet scheme has been included into the finite-difference anelastic model EULAG and its simplified version referred to as babyEULAG. EULAG and babyEULAG apply nonoscillatory-forward-in-time (NFT) integration scheme (e.g., Smolarkiewicz and Margolin, 1993; Grabowski and Smolarkiewicz, 2002; Prusa et al., 2008). For the coupling with super-droplets, the NFT scheme for the potential temperature (1) and water vapor mixing ratio (2) has been modified to include the Euler-forward time integration, that is,

$$\Psi(t + \Delta t) = [\Psi(t) + F(t)\,\Delta t]_0\,, \tag{6}$$

where $\Psi$ is either $\Theta$ or $q_v$, $F$ represents the right-hand-side of (1) and (2), and subscript "0" depicts the departure point of the fluid trajectory. This is the same as applied in the bin-microphysics versions of EULAG in Wyszogrodzki et al. (2011, section 2.2 therein) and babyEULAG in Grabowski and Jarecka (2015, appendix therein). Exploring the analogy between Lagrangian (trajectory-wise) and Eulerian (control-volume-wise) description of the fluid flow equations, (6) is solved using the flux-form monotone advection scheme MPDATA (e.g., Smolarkiewicz, 2006). Thus, the second-order-in-space and centered-in-time advection scheme is combined with the first-order-in-time (Euler forward) integration of the forcing term. Similar approach is used for the super-droplets, where the super-droplet transport is computed using the predictor-corrector scheme and droplet growth is calculated using the first-order-in-time un-centered scheme. It should be stressed the momentum equation in the host babyEULAG and EULAG models is advanced applying the centered in time scheme.

**2.3 Super-droplet initiation**

The key element of the scheme presented here that makes it distinct from the approach used in (Andrejczuk et al., 2008, 2010), Shima et al. (2009), Riechelmann et al. (2012), Arabas et al. (2015) and others, is the way super-droplets are created. The original implementations assume that super-droplets fill the entire computational domain, and they initially represent

[Figure]

Author: reviewer          Subject: Highlight          Date: 2017/10/08 21:17:49

9. Major request

To avoid confusion, the definition and time dependence of your multiplicity parameter has to be clarified. Based on the following discussion, I feel that regarding $N_i$ instead of $Q_{n,i} = N_i/\rho_a$ as the multiplicity parameter is the natural choice.

$N_i$, the number of RDs represented by the $i$-th SD, is constant in time under advection by the airflow. Admitting this, we can derive the number conservation eq. of the number density of RDs $n(\boldsymbol{x}, t)$ from the relationship (S1):

$$\partial_t n + \nabla \cdot (\boldsymbol{u} n) = 0. \tag{S2}$$

Therefore, as an attribute of SD, the number mixing ratio $Q_{n,i} = N_i/\rho_a$ is not constant in time. Let $Q_{n,i}(0) = N_i/\rho_a(0)$ be the initial number mixing ratio of the $i$-th SD. Then $Q_{n,i}(t) = N_i/\rho_a(t) = Q_{n,i}(0)(\rho_a(0)/\rho_a(t))$.

It is true that number mixing ratio is unchanged along a trajectory in Eulerian description. From (S2) and the mass conservation eq. $\partial_t \rho_a + \nabla \cdot (\boldsymbol{u}\rho_a) = 0$, we can derive that the number mixing ratio $q_n := n/\rho_a$ follows $\partial_t q_n + \boldsymbol{u} \cdot \nabla q_n = 0$. However, $Q_{n,i}$ is not constant due to the Lagrangian description. Note that the number density of SDs itself will be changed through advection (if the velocity field is not divergence free: $\nabla \cdot \boldsymbol{u} \neq 0$).

See also a simple example shown in the figure below. Here, $N_{\mathrm{RD}}$ and $N_{\mathrm{SD}}$ denote the number of RDs and SDs in the grid box.

[Figure]

$$N_{\mathrm{RD}}(t) = N_{\mathrm{RD}}(0)/2,$$
$$N_{\mathrm{SD}}(t) = N_{\mathrm{SD}}(0)/2,$$
$$\rho_a(t) = \rho_a(0)/2,$$
$$N_i(t) = N_i(0) = N_{\mathrm{RD}}/N_{\mathrm{SD}},$$
$$Q_{n,i}(t) = 2Q_{n,i}(0),$$
$$q_n(t) = q_n(0) = N_{\mathrm{RD}}/\rho_a/\Delta V.$$

deliquesced (humidified) CCN in equilibrium with their local environment. These un-activated super-droplets may become activated if environmental conditions dictate so, for instance, when passing through the cloud base. When CCN dry radius is one of the super-droplet attributes, the original approach allows explicit representation of aerosol processing by a cloud when collision/coalescence takes place (in which case the dry CCN after collision/coalescence combines dry CCN from col-

5  liding droplets; Shima et al., 2009) or when chemical reactions are included (e.g., A. Jaruga; PhD dissertation, University of Warsaw). However, if neither of those processes is of interest, a significantly simpler approach can be used based on the so-called Twomey activation (Twomey, 1959) as often used in bin microphysics schemes (e.g., Grabowski et al., 2011). Twomey approach links the number mixing ratio of activated CCN $N$ to the maximum supersaturation $S$ experienced by the cloudy volume. We will refer to the analytical or tabulated correspondence between $N$ and $S$ as the Twomey relationship. Cloud base

[revised manuscript text omitted]

Author: reviewer        Subject: Highlight        Date: 2017/10/08 21:21:20

11. Major request
Same as "9. Major request", please make it clear how the number mixing ratio of SD is tracked.

Author: reviewer        Subject: Highlight        Date: 2017/10/08 21:21:35

12. Comment
This is an important point worth to be mentioned.

At the same time, it is reported that applying equal multiplicity to SDs is not a good idea when collision/coalescence is considered. See, e.g., Unterstrasser et al. (2017). In short, very few but lucky big droplets could be important to explain the rain droplet size distribution development through collision/coalescence. However, equal multiplicity initialization could cause a sampling error problem, and results in a slow numerical convergence. See, e.g., Fig2c of Shima et al. (2009).

To satisfy both of these requirements, it would be better to assign small multiplicities to the SDs activated at very small supersaturation. (When collision/coalescence will be incorporated in the future.)

[Figure]

[Figure]

[Figure]

**Figure 2.** Illustration of the activation as represented on the fluid flow grid. Left panel shows locations of CCN activated at a given model time step. Right panel shows the situation at the next time step when activated CCN are advected away from the grid cell and activation of new CCN is required.

into the grid cell. Assuming that the supersaturation within the grid cell does not change, there is a need to activated new super-droplets as some of those present within the grid cell at the previous time step moved upwards. The new super-droplets should be introduced into the droplet-free volume (i.e., in the lower part of the grid cell in the right panel of Fig. 2) because un-activated CCN would be there. However, keeping track of volumes void of super-droplets during activation followed by advection is cumbersome. At the same time, adding new super-droplets randomly into the entire grid cell leads to the situation where super-droplets are not randomly distributed (i.e., more super-droplets is present in the upper part of the grid cell in Fig. 2). A simple approach adopted here is that all super-droplets are always randomly repositioned within a given grid cell once additional activation within that cell takes place.

**2.4 Transport of super-droplets in the physical space**

Super-droplets are advected in the physical space applying a predictor-corrector scheme to solve Eq. 5. The predictor step estimates the n+1 time level position from n time level velocity as:

$$x_p^{n+1} = x^n + u^n(x^n)\Delta t \,. \tag{7}$$

where the subscript "p" depicts the predictor solution. The corrector step (subscript "c") is subsequently applied as:

$$x_c^{n+1} = x^n + \left[ u^{n+1}(x_p^{n+1}) + u^n(x^n) \right] \frac{\Delta t}{2} \,. \tag{8}$$

The predictor-corrector scheme ensures the second-order accuracy for the time integration of the super-droplet transport. However, to increase accuracy, the corrector step can be repeated by replacing $x_p$ by already calculated $x_c$ in the $u^{n+1}$

[revised manuscript text omitted]

Author: reviewer          Subject: Highlight          Date: 2017/10/08 21:23:19

17. Minor request

How do the results depend on the SD advection time step dt? Perhaps dt = 1.0s is used for this test. The big error (of bi-linear and simple+predonly) should be reduced if we decrease dt. If you have any such data already, adding a few words on the dt dependece should be informative to the readers.

Author: reviewer          Subject: Highlight          Date: 2017/10/08 21:34:52

18. Major request

Please clarify at which timing the Twomey activation performed. Pehaps just after the transport of SDs?

Please also clarify when and how the cloud droplet deactivation (removal of SDs) is calculated. Perhaps when solving the droplet growth eq.(4)? And remove those SDs that have negative radius?

Author: reviewer          Subject: Highlight          Date: 2017/10/08 21:23:44

19. Typo

supersaturation

interpolation between a cloudy grid cell near the cloud edge and a sub-saturated cell outside the cloud results in sub-saturated conditions for super-droplets located near the cell boundary leading to their evaporation. In contrast, applying the mean supersaturation maintains the steady conditions near the motionless cloud-environment interface. Moreover, applying the Grabowski and Morrison (2008) methodology to cope with the spurious cloud-edge supersaturation discussed below becomes cumbersome (if not impossible) when the supersaturation interpolation to the super-droplet position is used. Overall, our tests similar to those discussed in the next section suggest that the impact of supersaturation interpolation in a rising thermal simulations is small and thus we decided to proceed with the simpler and computationally more efficient method of applying the grid-cell supersaturation to growth/evaporation of all super-droplets within a given grid cell.

**2.6 Avoiding spurious cloud-edge supersaturations**

The key aspect of the Grabowski and Morrison (2008) (GM08 hereinafter) method is to rely on the prediction of the absolute supersaturation (the difference between the water vapor mixing ratio and its saturated value) and to locally adjust the water vapor, cloud water, and temperature to maintain the predicted absolute supersaturation. This is in the spirit of Grabowski (1989) who used the temperature and supersaturation as main model variables and diagnosed the water vapor mixing ratio. Such a method results in a physically consistent supersaturation field but does not conserve water. GM08 circumvent this problem and apply the approach to the Eulerian double-moment cloud microphysics (i.e., predicting number and mass mixing ratios of the cloud water field). They also suggest how this approach can be used in the bin scheme (see section 4 therein). Here we explain how this method is used with Twomey super-droplets.

The crux of the method is to calculate the amount of cloud water $\epsilon$ that needs to condense or evaporate to ensure that the predicted potential temperature and water vapor mixing ratio fields give the absolute supersaturation that agrees with the predicted one. Thus, in addition to the prediction of the potential temperature and water vapor mixing ratio, the scheme predicts the evolution of the absolute supersaturation (see Eq. A8 in Grabowski and Morrison, 2008, and Eq. 4 in GM08). Once the amount of cloud water involved in the adjustment is calculated as in (7) of GM08, one needs to decide how that amount is distributed among super-droplets present within a given grid cell. Following GM08, the amount of cloud water $\epsilon$ that needs to be distributed among $N$ super-droplets from a given cell is calculated as

$$\epsilon = \sum_{i=1}^{N} \epsilon_i \,, \tag{11}$$

$$\epsilon_i = \frac{\epsilon}{\beta \tau_i} \,, \quad \text{where} \quad \beta = \sum_{k=1}^{N} \frac{1}{\tau_k} \,, \tag{12}$$

where $\tau_i$ is the phase relaxation time scale for the ith super-droplet (cf. A5 in Grabowski and Morrison, 2008). Knowing $\epsilon_i$, the radius of each super-droplet within a given grid cell is subsequently modified keeping the multiplicity parameter the same.
* * *
**Author: reviewer**    Subject: Highlight    Date: 2017/10/08 21:24:09

20. Typo
Morrison and Grabowski, 2008
* * *
**Author: reviewer**    Subject: Highlight    Date: 2017/10/08 21:24:13

21. Typo
italic
* * *
**Author: reviewer**    Subject: Highlight    Date: 2017/10/08 21:25:24

22. Major request
Please clarify the following points.

1) At which timing is this spurious supersaturation mitigation performed? Perhaps just after the transport of SDs, but before the Twomey activation?

2) In GM08, under sub-saturated condition, a limitation
epsilon <= 0
is imposed. Is this the same for Twomey SDs?

3) As explained in eq.6b of GM08, another limitation
epsilon >= -qc
is imposed for the bulk model because available cloud water is limited. In the same manner, is the limitation
epsilon_i >= - Ni mi / dV
imposed for Twomey SDs? If this condition is met, do you deactivate and remove the i-th SD from the list?
* * *
**Author: reviewer**    Subject: Highlight    Date: 2017/10/08 21:25:50

23. Minor request
Perhaps showing the exact form of tau_i  is helpful to the readers.
tau_i = 1/ {a2 (N_i/dV) r_i}
here dV is the volume of the grid.
* * *
**Author: reviewer**    Subject: Highlight    Date: 2017/10/08 21:25:59

24. Typo
Morrison and Grabowski, 2008

[Figure]

**3 Example of application: 2D moist thermal simulations**

[revised manuscript text omitted]

---

## Author Comment (AC1) · 14 Nov 2017

Responses to the Reviewer 1 comments. Original comments in italics, our responses in regular font.

*General comments:*

*The manuscript presents a new approach for the treatment of aerosol activation in Lagrangian Cloud Models (LCMs), a novel and promising approach for the simulation of cloud microphysics based on individually simulated super-droplets (SDs). The novelty (and advantage) of the new activation approach is that SDs are only introduced if the supersaturation exceeds a certain threshold. This is fundamentally different from previous activation approaches, in which SDs needed to be simulated even before activation. As pointed out in the manuscript, this new approach is not suited for simulating cloud- processing of aerosols. Applications in which cloud-processing of aerosols is of minor interest, however, benefit from reduced computing time as well as a smaller memory demand. Furthermore, the authors introduce further improvements necessary for the correct determination of supersaturations in LCMs: a velocity interpolation scheme which conserves the incompressibility of the flow as well as a technique to cope with spurious supersaturations. Since these latter refinements do not require the proposed aerosol activation scheme, they are a recommendable addition to all current LCMs.*
*All in all, this manuscript is well written, presents novel and useful methods, and is of interest to the entire LCM community. Accordingly, I recommend publishing this paper in Geoscientific Model Development. However, I would like the authors to address some minor comments, which will only increase the value of this already nice manuscript.*

We greatly appreciate the Reviewer's positive comments. We believe addressing the minor comments listed below resulted in a significantly improved presentation. The Reviewer's effort is greatly appreciated.

*Minor Comments*

*• p. 2, l. 25: Please define "multiplicity". It might be understandable but there are also synonyms used in the literature (e.g., weighting factor).*

We added a sentence with the definition.

*• p. 4, Eq. (4): There is a "+" missing between "ri" and "r0".*

Yes, we are sorry. That happened when we moved the text from a Word document we had been using when drafting the manuscript into the latex file. The error has been corrected.

*• sec. 2.3: Although the Twomey activation approach is new to warm-cloud LCMs, there is already an analog in ice-cloud LCMs. Sölch and Kärcher (2010) describe how they introduce new SIPs (simulation ice particles – the ice-cloud equivalent to SDs) to the model domain based on an underlying nucleation scheme, which exhibits many similarities to Twomey activation. Additionally, Unterstrasser and Sölch (2014) describe how a stochastic representation of that nucleation scheme can improve the model's statistics. I think these publications should be mentioned and discussed in the manuscript.*

Yes, we were aware of the similarity between ice initiation and Twomey activation of cloud droplets, but failed to point this out. This is corrected in the revised manuscript by bringing the Soelch and Kaercher reference. We do not feel that referring to the other paper is needed.

*• p. 6, ll. 11 – 12: Is the sentence "Without ... past." true if entrainment/mixing is considered? The diluted number mixing ratio cannot reveal the previous maximum supersaturation. (Although the Twomey activation scheme will still be applicable.)*

We believe the sentence in question is correct but it requires additional explanation. The diluted number mixing ratio represents the history of the supersaturation because it reflects a combination of past supersaturations in the two volumes: the undiluted cloudy volume and the cloud-free volume with un-activated aerosol. That said, we expect that details of a possible additional activation after an entrainment event might differ between the explicit activation scheme in the SD model and when the Twomey approach is used. This is especially true when CCN varies in the vertical. The same problem occurs in the Eulerian bin microphysics. We decided not to include the above discussion in the manuscript and we left the original text unmodified.

*• p. 7, ll. 12 – 14: I agree with the sentence "This is ... to another". However, the same multiplicity for all SDs might be disadvantageous for the initiation of collision and coalescence (see Unterstrasser et al., 2017).*

Yes, this is correct. We added a comment on that.

*• sec. 2.4: The suggested interpolation scheme should be used in all LCMs. However, there is one suggestion: Please add two plots to Fig. 4, which show the results for 100 SDs per grid cell, which is the typically applied SD concentration in current LCM simulations. This plot will be of great value to judge if there is a big impact of thoughtlessly applied tri-linear velocity interpolation in the published literature.*
*(Although I assume that there will be no impact visible due to the LCM's inherent fluctuations (now with a much higher standard deviation of around 10 %).)*

To address this point, we replaced the figure for the test with 1,000 SDs with a figure showing results for both 1,000 and 100 SDs. The new figure is included below. The discussion of the figure was modified, but the key point remains unchanged: bi-linear (or tri-linear in 3D) interpolation leads to unphysical concentration fluctuations when compared to the simple scheme.

[Figure]

• *p. 10, l. 29: Why is the calculation limited to cloudy regions (ql > 0.01 g kg-1)? Shouldn't the results be independent of the region within the model domain?*

This is just a small technicality. Passive particles were included only in part of the domain (as mentioned in line 20/21 of the original submission) to reduce the computational cost. If the entire domain is included in the calculation of the extrema, then the minimum would be zero, correct? Thus, we only use "cloudy points" defined using the specified threshold. We do not feel this needs to be dwelled upon in the text.

• *p. 14, ll. 19 – 21: Please give more details (or a reference) how the water condensation is split into 10 substeps. Based on the given text, I cannot imagine how this procedure is applied.*

Details of the sub-stepping in the UWLCM model are discussed in Arabas et al. (GMD 2015). We modified the sentence that mentions the sub-stepping (motivated also per Rev. 2 comment) and refer to Arabas et al. paper.

• *sec. 3.2: Please add some details on the number of SDs initialized in each grid box or the maximum number of SDs per grid box created by the Twomey activation scheme. These details follow later (p. 17, l. 6) but I expected them to be in the setup section.*

The text (with some modifications) was moved from page 17 to section 3.2. Also, per Rev. 2 request, we added information on the activation in the UWLCM as only part of SDs in the traditional approach become cloud droplets.

• *Fig. 9/10: How do you define activated particles? Using the Twomey activation scheme, this is a straightforward task. But how do you proceed in the UWLCM?*

Activated droplets in UWLCM are defined as those that have radius larger than the activation radius. This has been added to the text.

• *Fig. 9/10: Could you please comment a little more on the strong oscillations in the σ plots? It seems that those time series jump between two solutions.*

The oscillations are related to statistical fluctuations due to finite number of SDs as the amplitude is reduced roughly in proportion to the square root of SD number. It is important to note that the center of mass is calculated on the Eulerian grid, that is, it jumps from one grid box to another as the thermal moves upwards. The period of these oscillations in Figs. 9 and 10 (about 10 sec) matches the propagation of the center of mass over the grid (updraft velocity of about 2 m/s and grid length of 20 m). A comment on the has been added to the revised text.

• *p. 21, ll. 8 – 14: I agree that using the Twomey activation scheme will reduce the number of SDs in the model domain and, hence, computing time and memory demand. However, these considerations deserve some more thoughts. Models are usually parallelized using a 2D domain decomposition applied to the Eulerian fields but also the SDs. Accordingly, if there is a cloud in just in one subdomain, massive load imbalance will occur slowing down the whole computation. To benefit most of the new Twomey activation scheme, new parallelization strategies for the treatment of SDs need to be developed, e.g., a uniform distribution of SDs over all cores independent of their physical location in the model domain to avoid load imbalance issues.*

This is a valid point. However, the domain decomposition in physical space is used as a parallelization strategy in finite-difference models only. A parallelization for the Lagrangian thermodynamics should be developed outside of the domain decomposition and then load imbalances would not be an issue. We added a comment on that to the manuscript.

We addressed all technical comments through appropriate modifications of the text.

---

## Author Comment (AC2) · 14 Nov 2017

We sincerely appreciate the time and effort the Executive Editor took to point out drawbacks of our submission. The Editor points out that there are two requirements that our submission has not met. First, the paper title "must give the model name and version number (or other unique identifier) in the title". Second, all papers must include a section, at the end of the paper, entitled 'Code availability' and what we have in that section is not appropriate.

As for the first requirement, we would like to point out that our paper reports on the development of a novel methodology (Lagrangian condensation with Twomey activation) and not on the creation of a specific model or code. Thus, there is no version of the

code to be added to the title. Can a methodology or idea have a version number? Yes in principle, but we think this would be awkward. That said, we do include specifics of the codes used in testing the idea in the "Code availability" section as stated below. In addition, the manuscript discusses various specific aspects of the Lagrangian methodology (e.g., advection of super-droplets, avoiding spurious cloud-edge supersaturation, etc.) that are general and independent of the specific model or code. Although we understand the requirement, we feel that in our specific case such a requirement cannot be met. In fact, we looked at titles of manuscripts published or in review in GMD and we see that some of them (agreeably, the minority) do not have the version number in the title. Thus, we insist on the original title and hope our explanation is sufficient.

As for the second requirement, we agree with the Editor. This was our omission that will be corrected in the revised manuscript. The "Code availability" section will include links to the codes used in the simulations reported in the manuscript, including version numbers and dois.

---

## Author Comment (AC3) · 14 Nov 2017

Responses to the Reviewer 2 comments.

We greatly appreciate the Reviewer's effort in providing us with the annotated manuscript. This is very efficient way to communicate the comments. However, this also makes our job more difficult to communicate back our responses. We decided to pull out all comments marked as "major" from the annotated manuscript and respond to them in this document. The minor comments were either addressed by appropriate changes to the manuscript or are copied below together with our responses if no changes to the manuscript were done.

In the text below, the Reviewer's comments are in italics, our responses in regular font.

***Major comments:***

P. 5: *To avoid confusion, the definition and time dependence of your multiplicity parameter has to be clarified.*

Yes, we agree that the formulation in the Twomey activation formulation was in error. This was a mistake by the lead author that is now corrected. The code now uses multiplicity expressed as the number (i.e., the number of real droplets each super-droplet represents), not the number mixing ratio, in agreement with previous applications of the Lagrangian thermodynamics. After modifying the Twomey approach code the results are in a better agreement with the UWLCM. This is very much appreciated!

P. 12: *Please clarify at which timing the Twomey activation performed. Perhaps just after the transport of SDs? Please also clarify when and how the cloud droplet deactivation (removal of SDs) is calculated. Perhaps when solving the droplet growth eq. (4)? And remove those SDs that have negative radius?*

Activation takes place when the local supersaturation exceeds the "activation supersaturation", where the "activation supersaturation" is given by Twomey relationship applying the number mixing ratio for all droplets present in a given grid cell. This check is performed at every model time step applying updated thermodynamic Eulerian and Lagrangian variables, that is, as the final element of the time stepping algorithm (or the first element of the next time step). This allows additional temperature and moisture tendencies due to activation to be added to other forces that are derived at the same time and then used in the time-stepping procedure as given by Eq. 6 in the manuscript. At the same time, SDs are removed from the SD list when their radius becomes zero.

P. 13: *Please clarify the following points.*
*1) At which timing is this spurious supersaturation mitigation performed? Perhaps just after the transport of SDs, but before the Twomey activation?*
*2) In GM08, under sub-saturated condition, a limitation epsilon <= 0 is imposed. Is this the same for Twomey SDs?*

*3) As explained in eq.6b of GM08, another limitation epsilon >= -qc is imposed for the bulk model because available cloud water is limited. In the same manner, is the limitation epsilon_i >= - Ni mi / dV imposed for Twomey SDs? If this condition is met, do you deactivate and remove the i-th SD from the list?*

1) Since the model applies the Euler forward time integration scheme for thermodynamic fields (as explained in the manuscript, see Eq. 6), the adjustment takes place after calculations of new time level temperature and moisture variables are completed. Subsequently, the activation is calculated as it requires the final (i.e., adjusted so no spurious supersaturation values are present) thermodynamic variables within the grid cell.

2) Yes, an increase of the cloud water due to adjustment within a cell is not allowed if the air is sub-saturated. Please note that the sub-saturated air does not cause any harm in terms of droplet activation.

3) In the code, epsilon is first limited by the total liquid water as in the case of the bulk model. Then epsilon is "distributed" into epsilon_i for each SD and adjustment of each SD takes place. In the adjustment, the radius is changed and multiplicity is preserved as explained in the manuscript. And yes, the adjustment is allowed to remove all water from a given SD (such SD is later removed from the SD list). That said, we expect that complete elimination of a SD seldom happens as the amount of water substance involved in the adjustment is typically a small fraction of available water. This is our experience from 1D tests described in Grabowski and Morrison MWR manuscript and some limited testing we performed with the rising thermal simulations.

P. 14: *Let me confirm. Not in a per-grid manner, but in a per-particle manner? Does this mean that each SD does not feel the influence of other SDs in the same grid for 1sec? I suppose it is too long, at least to calculate activation/deactivation by solving the kappa-Kohler based growth equation.*

No, the "per-particle" term is confusing as thermodynamic fields are updated during the sub-stepping procedure. We revised the text removing the "per-particle" term and we refer to Arabas et al. (GMD 2015) that provides more detail.

**Minor comments that did not lead to significant changes and require an explanation:**

Abstract, L. 5-7: *Naumann and Seifert (2015, JAMES) applied the Lagrangian framework only to rain droplets. I think their work should be cited somewhere in the paper.*

We feel this is a very peculiar (but insightful as far as science is concerned) application of the Lagrangian methodology. We decided not to cite this paper. We removed the word "all" from the abstract.

P. 4, Eq. 4 and line 14: *Not the mass diffusivity Dv, but the thermal diffusivity \alpha should be used here. \alpha is about 15% smaller than Dv. See, e.g., table 2 of Montgomery (1947). Please update eq (4) or justify the use of Dv here.*

As explained in the manuscript, the simplified formulation (4) (except for the $r_0$ factor, i.e., $dr/dt = AS/r$) can be derived from the more accurate textbook formulation of the droplet growth equation that involves the sum of heat conduction and vapor diffusion factors (e.g., p. 73 in Rogers "Short course in cloud physics" book or the reference to Grabowski et al. 2011 paper given in the text). The derivation – again, as explained in the text – involves assuming that the thermal diffusivity is close to water diffusivity and using the latter in the thermal conductivity, $K = c_p \, \rho_a \, D_v$. The reviewer is correct, there is a 10-15% difference between the two that is neglected in the approximate formulation. Such a simplified formulation is used in the double-moment warm-rain scheme described in the appendix of Morrison and Grabowski (JAS 2008) referred to in the manuscript. We modified slightly the text to explicitly state the above.

---

## Author Response (AR1)

[revised manuscript text omitted]

---

## Author Response (AR2)

Additional comments by Topical Editor (in italics) and our responses (in regular font).

*p.6,l.18: please delete "perhaps"*

Removed.

*p.6,l.20: please mention the stochastic SD initialization of Unterstrasser & Sölch (2014)*

We do not think this is a correct place to cite the above paper. We included the citation in the place when our way of creating super-droplets is contrasted with other approaches.

*p.7,l.6: To be honest, I would rather say that "equal multiplicity is the WORST choice when collision/coalescence is ...".*

We revised the text to say "is possibly the worst choice".

*p.7,l.11: you write "there is a need to activated new super-droplets". Is it correct or did you mean "need for" or "need to activate"?*

This is a typo. The text should read "to activate". This has been corrected.

*p.10, last line: please replace "/" by "or" to make it clearer.*

Revised.

*The discussion in the last paragraph of section 3 must include the non-trivial point, that a SD memory re-organisation (as you propose) leads to increased inter-processor communication which could become a bottleneck in 2D/3D simulations.*

The Editor's claim is a speculation. Please think about the spectral DNS codes with droplets that do parallelization of the flow solver in the spectral space and then apply domain-decomposition in the physical space to move the droplets. We think one can design similar strategy for our approach, with the flow solver parallelized using domain decomposition (or any other strategy) and super-droplet calculations parallelized through dividing the super-droplet list among various processors. Whether such a strategy will lead to larger communication is an open issue. Overall, we think this is a caveat concerning implementation of our approach and we do not feel there is a need to speculate on this subject in the manuscript under review. We feel alerting the reader to the issue is sufficient at this stage.

*The Sölch & Kärcher reference should be added to your list of LCMs in the conclusion. Furthermore, I think it would be fair to mention there as well that this LCM introduced an analogous strategy much earlier. Whether the particular LCM treats warm clouds or pure cirrus is of secondary importance, as all mentioned LCM use similar concepts (even those they differ in some details of the implementations due to different underlying physics).*

The reference is again added to the conclusion section to follow the discussion in the main text.

[revised manuscript text omitted]